# Transient Neural Radiance Fields
# for Lidar View Synthesis and 3D Reconstruction

**Anagh Malik**[1,2]
anagh@cs.toronto.edu

**Parsa Mirdehghan**[1,2]
parsa@cs.toronto.edu

**Sotiris Nousias**[1]
sotiris@cs.toronto.edu

**Kiriakos N. Kutulakos**[1,2]
kyros@cs.toronto.edu

**David B. Lindell**[1,2]
lindell@cs.toronto.edu

[1]University of Toronto     [2]Vector Institute

## Abstract

Neural radiance fields (NeRFs) have become a ubiquitous tool for modeling scene appearance and geometry from multiview imagery. Recent work has also begun to explore how to use additional supervision from lidar or depth sensor measurements in the NeRF framework. However, previous lidar-supervised NeRFs focus on rendering conventional camera imagery and use lidar-derived point cloud data as auxiliary supervision; thus, they fail to incorporate the underlying image formation model of the lidar. Here, we propose a novel method for rendering *transient* NeRFs that take as input the raw, time-resolved photon count histograms measured by a single-photon lidar system, and we seek to render such histograms from novel views. Different from conventional NeRFs, the approach relies on a time-resolved version of the volume rendering equation to render the lidar measurements and capture transient light transport phenomena at picosecond timescales. We evaluate our method on a first-of-its-kind dataset of simulated and captured transient multiview scans from a prototype single-photon lidar. Overall, our work brings NeRFs to a new dimension of imaging at transient timescales, newly enabling rendering of transient imagery from novel views. Additionally, we show that our approach recovers improved geometry and conventional appearance compared to point cloud-based supervision when training on few input viewpoints. Transient NeRFs may be especially useful for applications which seek to simulate raw lidar measurements for downstream tasks in autonomous driving, robotics, and remote sensing.

## 1 Introduction

The ability to sense and reconstruct 3D appearance and geometry is critical to applications in vision, graphics, and beyond. Lidar sensors [1] are of particular interest for this task due to their high sensitivity to arriving photons and their extremely high temporal resolution; as such, they are being deployed in systems for 3D imaging in smart phone cameras [2], autonomous driving, and remote sensing [3]. Recent work has also begun to explore how additional supervision from lidar [4] or depth sensor measurements [5] can be incorporated into the NeRF framework to improve novel view synthesis and 3D reconstruction. Existing NeRF-based methods that use lidar [4] are limited to rendering *conventional RGB images*, and use lidar point clouds (i.e., pre-processed lidar measurements) as auxiliary supervision rather than rendering the raw data that lidar systems actually collect. Specifically, lidars capture *transient images*—time-resolved picosecond- or nanosecond-scale measurements of a pulse of light travelling to a scene point and back. We consider the problem of how to synthesize such transients from novel viewpoints. In particular, we seek a method that

37th Conference on Neural Information Processing Systems (NeurIPS 2023).

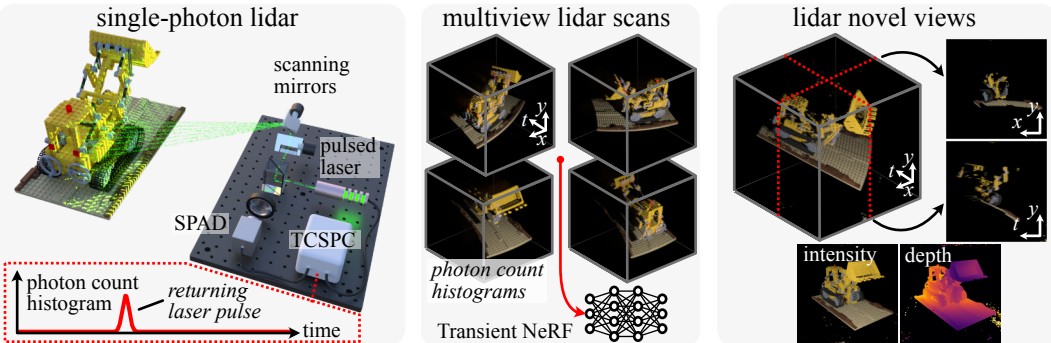

Figure 1: Overview of transient neural radiance fields (Transient NeRFs). Measurements from a single-photon lidar are captured using a single-photon avalanche diode (SPAD), pulsed laser, scanning mirrors, and a time-correlated single photon counter (TCSPC). The lidar scans, consisting of a 2D array of photon count histograms (visualized with maximum-intensity projection), are captured from multiple viewpoints and used to optimize the transient NeRF. After training, we render novel views of time-resolved lidar measurements ($x$–$y$ and $x$–$t$ slices are indicated by the dotted red lines), and we also convert the rendered data into intensity and depth maps.

takes as input and renders transients in the form of time-resolved photon count histograms captured by a single-photon lidar system[1] [8]. Lidar view synthesis may be useful for applications that seek to simulate raw lidar measurements for downstream tasks, including autonomous driving, robotics, remote sensing, and virtual reality.

The acquisition and reconstruction of transient measurements has been studied across various different sensing modalities, including holography [9], photonic mixer devices [10, 11] streak cameras [12], and single-photon detectors (SPADs) [13, 14].

In the context of SPADs and single-photon lidar, a transient is measured by repeatedly illuminating a point with pulses of light and accumulating the individual photon arrival times into a time-resolved histogram. After capturing such histograms for each point in a scene, one can exploit their rich spatio-temporal structure for scene reconstruction [15, 16], to uncover statistical properties of captured photons [17, 18], and to reveal the temporal profile of the laser pulse used to probe the scene (knowledge of which can significantly improve depth resolution [19, 20]). These properties motivate transients as a representation and their synthesis from novel views. While existing methods have explored multiview lidar reconstruction [21–25], they exclusively use point cloud data, and do not tackle lidar view synthesis.

Recently, a number of NeRF-based methods for 3D scene modeling have also been proposed to incorporate point cloud data (e.g., from lidar or structure from motion) [26, 27] or information from time-of-flight sensors [5]. Again, these methods focus on synthesizing conventional RGB images or depth maps, while our approach synthesizes transient images. Another class of methods combines NeRFs with single-photon lidar data for non-line-of-sight imaging [28]; however, they focus on a very different inverse problem and scene parameterization [29], and do not aim to perform novel view synthesis of lidar data as we do.

Our approach, illustrated in Fig. 1, extends neural radiance fields to be compatible with a statistical model of time-resolved measurements captured by a single-photon lidar system. The method takes as input multiview scans from a single-photon lidar and, after training, enables rendering lidar measurements from novel views. Moreover, accurate depth maps or intensity images can also be rendered from the learned representation.

In summary, we make the following contributions.

- We develop a novel time-resolved volumetric image formation model for single-photon lidar and introduce transient neural radiance fields for lidar view synthesis and 3D reconstruction.

---

[1]Single-photon lidars are closely related to conventional lidar systems based on avalanche photodiodes [6], but they have improved sensitivity and timing resolution (discussed in Section 2); other types of coherent lidar systems [7] are outside the scope considered here.

- We assemble a first-of-its-kind dataset of simulated and captured transient multiview scans, constructed using a prototype multiview single-photon lidar system.
- We use the dataset to demonstrate new capabilities in transient view synthesis and state-of-the-art results on 3D reconstruction and appearance modeling from few (2–5) single-photon lidar scans of a scene.

## 2   Related work

Our work ties together threads from multiple areas of previous research, including methods for imaging with single-photon sensors, and NeRF-based pipelines that leverage 3D information to improve reconstruction quality. Our implementation also builds on recent frameworks that improve the computational efficiency of NeRF training [30, 31].

**Active single-photon imaging.**   Single-photon sensors output precise timestamps corresponding to the arrival times of individual detected photons. The most common type of single-photon sensor is the single-photon avalanche diode (SPAD). SPADs are based on the widely-available CMOS technology [32] (which we consider in this work), but other technologies such as superconducting nanowire single-photon detectors [33] and silicon photomultipliers [34], offer different tradeoffs in terms of sensitivity, temporal resolution, and cost.

In active imaging scenarios, pulsed light sources are paired with single-photon sensors to estimate the depth or reflectance of a scene by applying computational algorithms to the captured photon timestamps [19, 35, 36]. The extreme temporal resolution of these sensors also enables direct capture of interactions of light with a scene at picosecond timescales [37, 38], and by modeling and inverting the time-resolved scattering of light, single-photon sensors can be used to see around corners [28, 39–41] or through scattering media [42–44]. The extreme sensitivity of single-photon sensors has made them an attractive technology for autonomous navigation [8], and accurate depth acquisition from mobile phones [2].

Our approach differs significantly from all the previous work in that we investigate, for the first time, the problem of lidar view synthesis and multi-view 3D reconstruction in the single-photon lidar regime. We introduce the framework of transient NeRFs for this task and jointly optimize a representation of scene geometry and appearance that is consistent with captured photon timestamps across all input views.

Finally, we note that while commercial lidars based on SPADs or avalanche photodiodes capture photon count histograms or time-resolved intensity, they typically pre-process these raw measurements to point cloud format before output. Hence, the raw lidar data that we use may not have been widely available to previous methods; we have publicly released our dataset of multiview photon count histograms on the project webpage to help alleviate this challenge.

**3D-informed neural radiance fields.**   A number of recent techniques for multiview reconstruction using NeRFs leverage additional geometric information (sparse point clouds from lidar [4] or structure from motion [26, 27]) to improve the reconstruction quality or reduce the number of required input viewpoints.   Similar benefits can be obtained by combining volume rendering with monocular depth estimators [45], or using data from time-of-flight sensors [5]. Other methods investigate the problem of view synthesis from few input images but leverage appearance priors instead of explicit depth supervision [46–48].   In contrast to the proposed approach, all of these methods focus on reconstructing images or depth maps rather than transients.

## 3   Transient Neural Radiance Fields

We describe a mathematical model for transient measurements captured using single-photon lidar and propose a time-resolved volume rendering formulation compatible with neural radiance fields.

### 3.1   Image Formation Model

Consider that a laser pulse illuminates a point in a scene that is imaged onto a sensor at position $\mathbf{p} \in \mathbb{R}^2$ (see Fig. 2). Assume light from the laser pulse propagates to a surface and back to $\mathbf{p}$ along

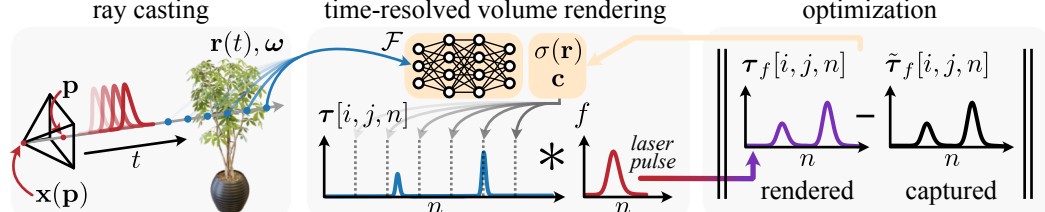

Figure 2: Rendering transient neural radiance fields. We cast rays through a volume and retrieve the density and color at each point using a neural representation [30]. A time-resolved measurement is constructed using volume rendering (Equation 3), and we bin the radiance contributions into an array based on distance along the ray. The result is convolved with the impulse response of the lidar (which incorporates the shape of the laser pulse), and we supervise the neural representation based on the difference between the rendered and captured transient measurements.

the same path described by a ray $\mathbf{r}(t)$, where $t$ indicates propagation time. The forward path along the ray is given as $\mathbf{r}(t) = \mathbf{x}(\mathbf{p}) + tc\,\boldsymbol{\omega}(\mathbf{p})$, where $\mathbf{x}(\mathbf{p}) \in \mathbb{R}^3$ is the ray origin, $\boldsymbol{\omega}(\mathbf{p}) \in \mathbb{S}^2$ is the ray direction which maps to $\mathbf{p}$, and $c$ is the speed of light. Now, let $f(t)$ denote the temporal impulse response of the lidar (including the temporal profile of the laser pulse and the sensor jitter), and let $\alpha(\mathbf{p})$ incorporate reflectance and radiometric falloff factors [17] of the illuminated point at distance $z(\mathbf{p})$ from $\mathbf{x}(\mathbf{p})$. Then, assuming single-bounce light transport, the photon arrival rate incident on the sensor from the laser pulse is given as

$$\boldsymbol{\lambda}[i,j,n] = \int_{\mathcal{P}_{i,j}} \int_{\mathcal{T}_n} \alpha(\mathbf{p})\, f\left(t - \frac{2z(\mathbf{p})}{c}\right)\, \mathrm{d}t\, \mathrm{d}\mathbf{p}, \tag{1}$$

where $\mathcal{T}_n$ and $\mathcal{P}_{i,j}$ indicate the temporal and spatial discretization intervals for the time bin $n$ and pixel $i, j$, respectively. The term $2z/c$ gives the time delay for light to propagate to a point at distance $z$ and back.

Now, we can describe the measured transient, or the number of photon detections captured by a SPAD [17], as

$$\widetilde{\boldsymbol{\tau}}[i,j,n] \sim \text{POISSON}\left(N\eta\,\boldsymbol{\lambda}[i,j,n] + B\right), \quad B = N(\eta A[i,j] + D), \tag{2}$$

where $N$ indicates the number of laser pulses per pixel, $\eta \in (0,1)$ is the detection efficiency of the sensor, and $B$ is the total number of background (non-laser pulse) detections. Background detections in turn depend on $A$, the average ambient photon rate at pixel $[i,j]$, and $D$, number of false detections produced by the sensor per laser pulse period, also known as the dark count rate. When the number of detected photons is far fewer than the number of laser pulses, SPAD measurements can be modeled according to a Poisson process [17] where the arrival rate function varies across space and time. This model is appropriate for our measurements, which have relatively low flux ($< 5\%$ detection probability per emitted laser pulse) [19]. The resulting measurements $\widetilde{\boldsymbol{\tau}}[i,j,n]$ represent a noisy histogram of photon counts collected at pixel $[i,j]$ at time bin $n$.

## 3.2 Time-Resolved Volume Rendering

Using the measurements $\widetilde{\boldsymbol{\tau}}$, we wish to optimize a representation of the appearance and geometry of the scene. To this end, we propose a time-resolved version of the volume rendering equation used in NeRF [49, 50]. Specifically, we model clean (i.e., without Poisson noise) time-resolved histograms $\boldsymbol{\tau}[i,j,n]$ as (writing $\mathbf{r}(t)$ as $\mathbf{r}$ for brevity)

$$\boldsymbol{\tau}[i,j,n] = \int_{\mathcal{P}_{i,j}} \int_{\mathcal{T}_n} (tc)^{-2}\, T(t)^2 \sigma(\mathbf{r}) \mathbf{c}(\mathbf{r}, \boldsymbol{\omega})\, \mathrm{d}t\, \mathrm{d}\mathbf{p},$$

$$\text{where } T(t) = \exp\left(-\int_{t_0}^{t} \sigma(\mathbf{r})\, \mathrm{d}s\right). \tag{3}$$

We denote by $\mathbf{c}$ the radiance of light scattered at a point $\mathbf{r}(t)$ in the direction $\boldsymbol{\omega}$, and $\sigma$ represents the volume density or the differential probability of ray termination at $\mathbf{r}(t)$. Finally, $T(t)$ is the

transmittance from a distance $t_0$ along the ray to $t$, and this term is squared to reflect the two-way propagation of light [5]. We additionally explicitly account for the inverse-square falloff of intensity, through the term $(tc)^{-2}$. The definite integrals are evaluated over the extent of time bin $n$, $\mathcal{T}_n = [t_{n-1}, t_n]$, and over $\mathbf{p}$ within the area of pixel $[i, j]$ as in Equation 1. Note that in practice, we calculate Equation 1 using the discretization scheme of Max [51] used by Mildenhall et al. [49].

Finally, to account for the temporal spread of the laser pulse and sensor jitter, we convolve the estimated transient with the calibrated impulse response of the lidar system $f$ to obtain

$$\boldsymbol{\tau}_f = f * \boldsymbol{\tau}. \tag{4}$$

Without this step, the volumetric model of Equation 3 does not match the raw data from the lidar system and tends to produce thick clouds of density around surfaces to compensate for this mismatch.

### 3.3  Reconstruction

To reconstruct transient NeRFs, we use lidar measurements $\widetilde{\boldsymbol{\tau}}^{(k)}[i, j, n]$ of a scene captured from $0 \leq k \leq K - 1$ different viewpoints. We parameterize transient NeRF using a neural network $\mathcal{F}$ consisting of a hash grid of features and a multi-layer perceptron decoder [30]. The network takes as input a coordinate and viewing direction, and outputs radiance and density, $\mathcal{F}(\mathbf{r}(t), \boldsymbol{\omega}) = \mathbf{c}, \sigma$. We use these outputs to render transients (see Fig. 2). The model is optimized to minimize the difference between the rendered transient and measured photon count histograms. We also introduce a modified loss function to account for the high dynamic range of lidar measurements, and we propose a space carving regularization penalty to help mitigate convergence to local minima in the optimization.

**HDR-Informed loss function.**   Measurements from a single photon sensor can have a dynamic range that spans multiple orders of magnitude, with each pixel recording from zero to thousands of photons. We find that applying two exponential functions to the radiance preactivations (1) enforces non-negativity and (2) improves the dynamic range of the network output. Thus, we have $\mathbf{c} = \exp(\exp(\hat{\mathbf{c}})) - 1$, where the network preactivations are given by $\hat{\mathbf{c}}$. Following Muller et al. [30] the network also predicts density in log space.

After time-resolved volume rendering using Equation 3, we apply a loss function in log space to prevent the brightest regions from dominating the loss [52]. The loss function is given as

$$\mathcal{L}_{\boldsymbol{\tau}} = \sum_{k, i, j, n} \| \ln(\widetilde{\boldsymbol{\tau}}^{(k)}[i, j, n] + 1) - \ln(\boldsymbol{\tau}_f^{(k)}[i, j, n] + 1) \|_1, \tag{5}$$

where the sum is over all images, pixels, and time bins.

**Space carving regularization.**   We find that using the above loss function alone results in spurious patches of density in front of dark surfaces in a scene. Here, the network can predict bright values on the surface itself, but darkens the corresponding values of $\boldsymbol{\tau}_f$ by placing additional spurious density values along the ray. Since the network can predict the radiance of the density to be zero at these points, the predicted transients $\boldsymbol{\tau}_f$ can be entirely consistent with the measured transients $\widetilde{\boldsymbol{\tau}}$, but with incorrect geometry. To address this, we introduce a space carving regularization

$$\mathcal{L}_{\text{sc}} = \sum_{\substack{k, i, j, n \\ \widetilde{\boldsymbol{\tau}}^{(k)}[i, j, n] < B}} \int_{\mathcal{P}_{i,j}} \int_{\mathcal{T}_n} T(t)\sigma(\mathbf{r}) \, dt \, d\mathbf{p}. \tag{6}$$

This function penalizes any density along a ray at locations where the corresponding measured transient values are less than the expected background level $B$. This effectively forces space to be empty (i.e., zero density) at regions where the measurements do not indicate the presence of a surface.

The complete loss function used for training is then given as

$$\mathcal{L} = \mathcal{L}_{\boldsymbol{\tau}} + \lambda_{\text{sc}} \mathcal{L}_{\text{sc}}, \tag{7}$$

where $\lambda_{\text{sc}}$ controls the strength of the space carving regularization.

### 3.4 Implementation Details

Our implementation is based on the NerfAcc [31] version of Instant-NGP [30], which we extend to incorporate our time-resolved volume rendering equation. In particular, we extend the framework to output time-resolved transient measurements, to account for the pixel footprint, and to estimate depth.

**Pixel footprint.** Captured photon count histograms exhibit multiple peaks where the finite beam width of the laser passes over depth discontinuities. To account for this phenomenon we use a truncated Gaussian distribution to model the spatial footprint of the laser spot and SPAD sensor projected onto the scene. Specifically, we sample rays in the range of 4 standard deviations of the pixel center, weighting their contribution to the rendered transient by the corresponding Gaussian probability density function value (after rendering a histogram). We set the standard deviation of the Gaussian to 0.15 pixels for the simulated dataset and 0.10 pixels for the captured dataset.

**Depth.** To estimate depth we find the distance along each ray that results in the maximum probability of ray termination: $\mathrm{argmax}_t \, T(t)\sigma(t)$. Note that when integrating over the pixel footprint at occlusion boundaries, multiple local extrema can occur, and so taking the highest peak results in a single depth estimate without floating pixel artifacts.

**Network optimization.** We optimize the network using the Adam optimizer [53], a learning rate of $1 \times 10^{-3}$ and a multi-step learning rate decay of $\gamma = 0.33$ applied at 100K, 150K, and 180K iterations. We set the batch size to 512 pixels and optimize the simulated results until they appear to converge, or for 250K iterations for the simulated results and 150K iterations for the captured results. For the weighting of the space carving loss, we use $\lambda_{sc} = 10^{-3}$ for the simulated dataset and increase this to $\lambda_{sc} = 10^{-2}$ for captured data, which benefits from additional regularization. We train the network on a single NVIDIA A40 GPU. Also note that simulated results use RGB histograms, such that $\mathbf{c} \in \mathbb{R}^3$. In the captured data $\mathbf{c} \in \mathbb{R}$ because we illuminate the scene with monochromatic laser light.

## 4 Multiview Lidar Dataset

We introduce a first-of-its-kind dataset consisting of simulated and captured multiview data from a single-photon lidar. A description of the full set of simulated and captured scenes is included in the supplemental, and the dataset and simulation code are publicly available on the project webpage.

**Simulated dataset** We create the simulated dataset using a time-resolved version of Mitsuba 2 [54] which we modify for efficient rendering of lidar measurements. The dataset consists of one scene from Vicini et al. [55] and four scenes made available by artists on Blendswap (`https://blendswap.com/`), which we ported from Blender to our Mitsuba 2 renderer. The training views are set consistent with the capture setup of our hardware prototype (described below) such that the camera viewpoint is rotated around the scene at a fixed distance and elevation angle, resulting in 8 synthetic lidar scans used for training. We evaluate on rendered measurements from six viewpoints sampled from the NeRF Blender test set [49]. The renders are used to simulate SPAD measurements by applying the noise model described in Equation 2 and setting the mean number of photon counts to 2850 per occupied pixel and the background counts to 0.001 per bin, which we set to approximate our experimentally captured data.

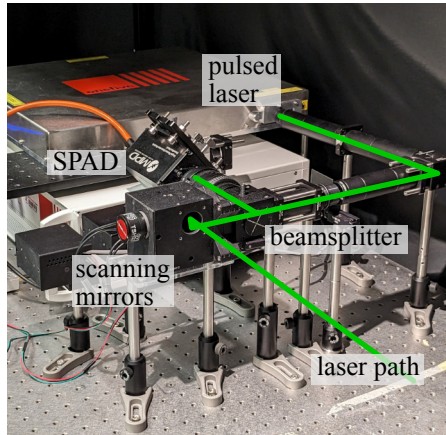

Figure 3: Hardware prototype. A pulsed laser shares a path with a single-pixel SPAD, and the illumination and imaging path are controlled by scanning mirrors.

**Hardware prototype.** To create the captured dataset, we built a hardware prototype (Fig. 3) consisting of a pulsed laser operating at 532 nm that emits 35 ps pulses of light at a repetition rate of 10 MHz. The output power of the laser is lowered to $< 1$ mW to keep the flux low enough

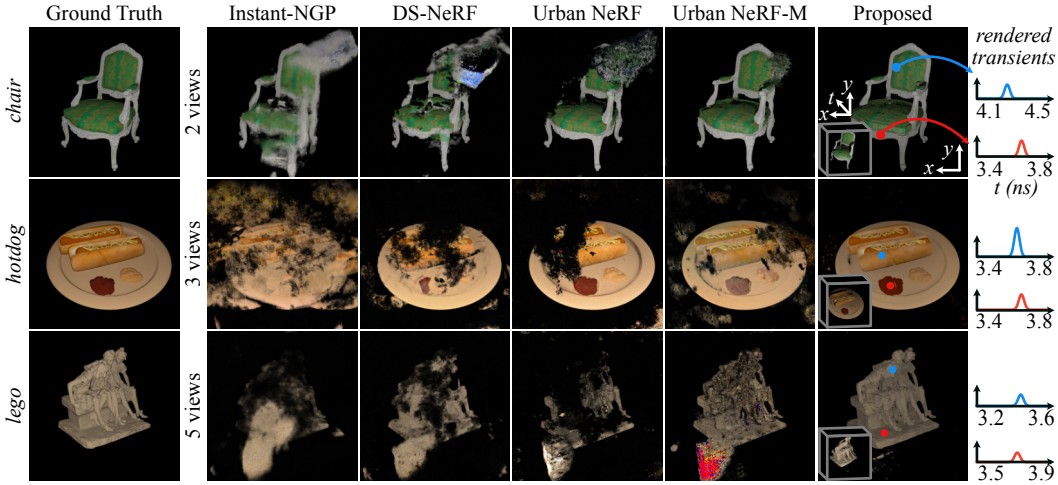

Figure 4: Results on simulated data. We show images from depth-supervised NeRF baselines as well as color images and rendered transients from our method after training on 2, 3, and 5 viewpoints. The proposed method produces cleaner results and generates 3D transients for each viewpoint.

Table 1: Simulated results comparing images and depth for the baselines and proposed approach.

| Method | PSNR (dB)↑ | | | LPIPS↓ | | | L1 (depth)↓ | | |
|---|---|---|---|---|---|---|---|---|---|
| | 2 views | 3 views | 5 views | 2 views | 3 views | 5 views | 2 views | 3 views | 5 views |
| Instant NGP [30] | 16.62 | 17.67 | 19.66 | 0.520 | 0.476 | 0.387 | 0.238 | 0.195 | 0.178 |
| DS-NeRF [26] | 19.28 | 19.35 | 21.07 | 0.431 | 0.436 | 0.376 | 0.109 | 0.115 | 0.119 |
| Urban NeRF [4] | 18.86 | 18.73 | 19.80 | 0.500 | 0.484 | 0.406 | 0.131 | 0.124 | 0.101 |
| Urban NeRF w/mask [4] | 20.91 | 20.81 | 22.34 | 0.410 | 0.382 | 0.339 | 0.051 | 0.038 | 0.029 |
| Proposed | 21.38 | 23.48 | 28.39 | 0.172 | 0.151 | 0.115 | 0.015 | 0.011 | 0.013 |

(roughly $150,000$ counts per second on average) to prevent pileup, which is a non-linear effect that distorts the SPAD measurements [56]. The laser shares an optical path with a single-pixel SPAD through a beamsplitter, and a set of 2D scanning mirrors is used to raster scan the scene at a resolution of $512{\times}512$ scanpoints. A time-correlated single-photon counter is used to record the photon timestamps with a total system resolution of approximately 70 ps.

**Captured dataset.** We capture multiview lidar scans of six scenes by placing objects on a rotation stage in front of the scanning single-photon lidar and capturing 20 different views in increments of 18 degrees of rotation. For each lidar scan we accumulate photons during a 20 minute exposure time to minimize noise in the transient measurements. We bin the photon counts into histograms with 1500 bins and bin widths of 8 ps (all raw timestamp data will also be made available with the dataset). We set aside 10 views sampled in 36 degree increments for testing and we use 8 of the remaining views for training. Prior to input into the network for training, we normalize the measurement values by the maximum photon count observed across all views.

**Calibration.** We calibrate the camera intrinsics of the system using a raxel model [57] with corners detected from two scans of checkerboard translated in a direction parallel to the surface normal. This model calibrates the direction of each ray individually, which is necessary because the 2D scanning mirrors deviate from the standard perspective projection model [58]. Extrinsics are calibrated by placing a checkerboard on a rotation stage and solving for the axis and center of rotation that best align the 3D positions of the checkerboard corners, where the 3D points are found using the calibrated ray model along with the time of flight from the lidar (see supplemental). Overall, accurate calibration is an important and non-trivial task because multiview lidar scans provide two distinct geometric constraints (i.e. stereo disparity and time of flight) that must be consistent for scene reconstruction.

# 5  Results

We evaluate our method on the simulated and captured datasets and use transient neural radiance fields to render intensity, depth, and time-resolved lidar measurements from novel views.

**Baselines.**   The intensity and depth rendered from our method are compared to four other baseline methods that combine point cloud-based losses with neural radiance fields. For fairer comparison and to speed up training and inference times, we implement the baselines by incorporating their loss terms into the recently introduced frameworks of NerfAcc [31] and Instant-NGP [30] adopted by our method. We train the following baselines using intensity images (i.e., the photon count histograms integrated over time) along with point clouds obtained from the photon count histograms using a log-matched filter, which is the constrained maximum likelihood depth estimate [59].

- Instant-NGP [30] is used to illustrate performance without additional depth supervision.
- Depth-Supervised NeRF (DS-NeRF) [26] incorporates an additional loss term to ensure that the expected ray termination distance in volume rendering aligns with the point cloud points.
- Urban NeRF [4] incorporates the ray-termination loss of DS-NeRF while also adding space carving losses to penalize density along rays before and after the intersection with a point cloud point.
- Urban NeRF with masking (Urban NeRF-M) modifies Urban NeRF to incorporate an oracle object mask and extends the space carving loss to unmasked regions, providing stronger geometry regularization (additional details in supplement).

Prior to input into the network, we normalize the images and apply a gamma correction, which improves network fitting to the high dynamic range data. Finally, after training with each method, we estimate an associated depth map using the expected ray termination depth at each pixel, which is the same metric used in the loss functions of the aforementioned baselines.

## 5.1  Simulated Results

The method is compared to the baselines in simulation across five scenes: *chair*, *ficus*, *lego*, *hot dog*, and *statue*. In Fig. 4, we show RGB images rendered from novel views using the baselines and our proposed method after training on two, three, and five views. More extensive sets of results on all scenes are included in the supplemental. We find that views rendered from transient neural radiance fields have fewer artifacts and spurious patches of density, as the explicit supervision from the photon count histograms avoids the ill-posedness of the conventional multiview reconstruction problem.

Additional quantitative results are included in Table 1, averaged across all simulated scenes. For the evaluation of rendered RGB images, we normalize and gamma-correct the output of the proposed method and the ground truth in the same fashion as the baseline methods. Transient NeRF recovers novel views with significantly higher peak signal-to-noise ratio and better performance on the learned perceptual image patch similarity (LPIPS) metric [60] compared to baselines. Transient measurements provide explicit supervision of the unoccupied spaces in the scene, leading to fewer floating artifacts, and cleaner novel views.

The depth maps inferred from Transient NeRF are also significantly more accurate than baselines (see Fig. 5). One key advantage here is that we avoid supervision on point clouds obtained by potentially

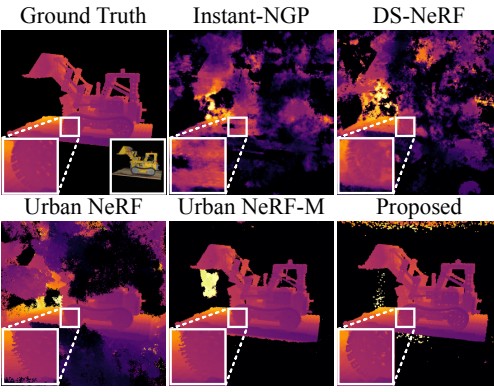

Figure 5: Comparison of depth maps recovered from simulated measurements trained on 5 views of the *lego* scene.

noisy (and thus view-inconsistent) per-pixel estimates of depth. By training on the raw photon count histograms, the scene's geometry is allowed to converge to the shape that best explains all histograms across all views, resulting in much higher geometric accuracy.

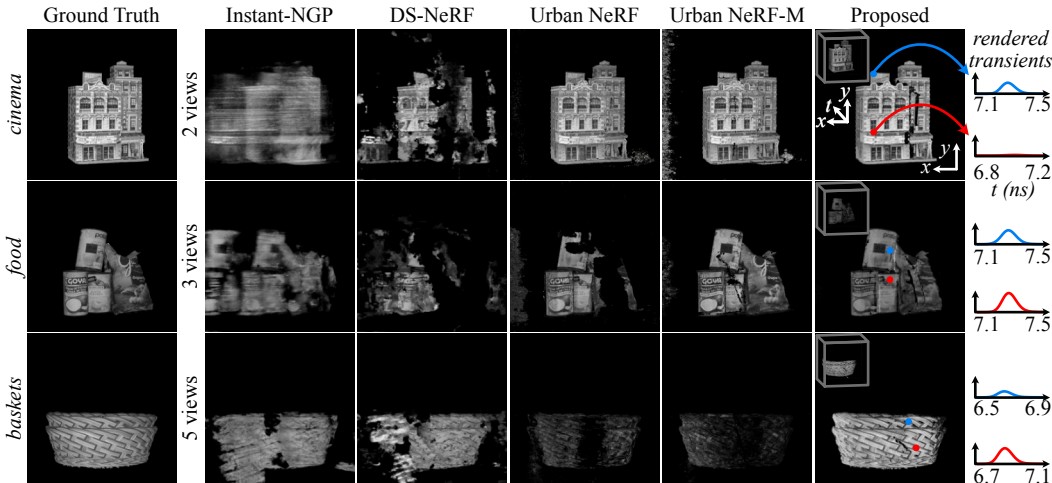

Figure 6: Results on multiview lidar data captured with the hardware prototype and trained with 2, 3, and 5 viewpoints. For the proposed method we show the rendered transients, intensity image, and individual transients for the indicated pixels.

Table 2: Evaluation of rendered intensity images and depth on captured results.

| Method | PSNR (dB) ↑ | | | LPIPS ↓ | | | L1 (depth) ↓ | | |
|---|---|---|---|---|---|---|---|---|---|
| | 2 views | 3 views | 5 views | 2 views | 3 views | 5 views | 2 views | 3 views | 5 views |
| Instant NGP [30] | 16.44 | 16.52 | 16.39 | 0.358 | 0.307 | 0.274 | 0.115 | 0.076 | 0.053 |
| DS-NeRF [26] | 15.34 | 15.05 | 14.86 | 0.311 | 0.312 | 0.325 | 0.048 | 0.036 | 0.036 |
| Urban NeRF [4] | 16.90 | 15.91 | 15.93 | 0.403 | 0.328 | 0.231 | 0.017 | 0.015 | 0.014 |
| Urban NeRF w/mask [4] | 15.45 | 18.26 | 19.11 | 0.458 | 0.269 | 0.191 | 0.014 | 0.006 | 0.004 |
| Proposed | 22.11 | 21.83 | 22.72 | 0.271 | 0.212 | 0.172 | 0.005 | 0.006 | 0.010 |

## 5.2 Captured Results

In Fig. 6 we show rendered novel views of intensity images from our method and baselines trained on captured data with two, three, and five views. Results are shown on the *cinema*, *food*, and *baskets* scenes (additional results in the supplemental). The proposed approach results in fewer artifacts and the rendered intensity images are more faithful to reference intensity images captured from the novel viewpoint. Quantitative comparisons of our method to baselines on captured data are shown in Table 2; note that we do not have access to ground truth depth for captured data and instead use depth from a log-matched filter on the ground truth transient. We find that the method outperforms the baselines in terms of PSNR and LPIPS of intensity images rendered from novel views. While performance on captured data does not improve as much as observed on simulated data with increasing numbers of viewpoints, we attribute this to small imperfections (≈1 mm) in the alignment of the lidar scans after estimating the camera extrinsics.

Since DS-NeRF trains explicitly on depth without additional regularization, it is especially sensitive to camera perturbations and can be outperformed in some cases by Instant NGP which has no additional geometry constraints. Our approach appears somewhat less sensitive to these issues, perhaps because geometry regularization is done implicitly through a photometric loss on the lidar measurements.

We notice some degradation in depth accuracy relative to simulation, likely due to imperfections in the estimated extrinsics. Sub-mm registration of the lidar measure-

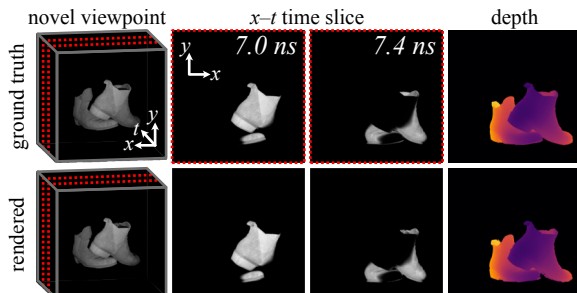

Figure 7: Comparison between reference and novel views of lidar measurements, intensity slices, and depth. The $x$–$y$ intensity slices are visualized for times indicated by the red dashed lines.

ments would likely improve results, but achieving such precise registration is non-trivial and beyond the scope of our current work. Finally, in Fig. 7 we compare captured measurements to rendered transients and depth rendered for the *boots* scene trained on 2 viewpoints. We recover the time-resolved light propagation from a novel view, shown in $x$–$y$ slices of the rendered transients over time. The depth map recovered from the novel view appears qualitatively similar to the ground truth (estimated from captured measurements using a log-matched filter [17]). We show additional 3D reconstruction results in the supplemental.

## 6    Discussion

Our work brings NeRF to a new dimension of imaging at transient timescales, offering new opportunities for view synthesis and 3D reconstruction from multiview lidar. While our work is limited to modeling the direct reflection of laser light to perform lidar view synthesis, our dataset captures much richer light transport effects, including multiple bounces of light and surface reflectance properties that could open avenues for future work. In particular, the method and dataset may help enable techniques for intra-scene non-line-of-sight imaging [61–64] (i.e., recovering geometry around occluders within a scene), and recovery of the bidirectional reflectance distribution function via probing with lidar measurements [65].

Our method is also limited in that we do not explore more view synthesis in more general single-photon imaging setups, such as when the lidar and SPAD are not coaxial; we hope to explore these configurations in future work. The proposed framework and the ability to render transient measurements from novel views may be especially relevant for realistic simulation for autonomous vehicle navigation, multiview remote sensing, and view synthesis of more general transient phenomena.

## Acknowledgments and Disclosure of Funding

Kiriakos N. Kutulakos acknowledges the support of the Natural Sciences and Engineering Council of Canada (NSERC) under the RGPIN and RTI programs. David B. Lindell acknowledges the support of the NSERC RGPIN program. The authors also acknowledge Gordon Wetzstein and the Stanford Computational Imaging Lab for loaning the single-photon lidar equipment.

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
