# Transient Neural Radiance Fields
# for Lidar View Synthesis and 3D Reconstruction
# –Supplementary Material–

**Anagh Malik**[1,2]
anagh@cs.toronto.edu

**Parsa Mirdehghan**[1,2]
parsa@cs.toronto.edu

**Sotiris Nousias**[1]
sotiris@cs.toronto.edu

**Kiriakos N. Kutulakos**[1,2]
kyros@cs.toronto.edu

**David B. Lindell**[1,2]
lindell@cs.toronto.edu

[1]University of Toronto    [2]Vector Institute

## 1 Hardware Prototype

To create the captured dataset, we built a hardware prototype consisting of a pulsed laser (NKT Photonics Katana 05HP) operating at 532 nm that emits 35 ps pulses of light at a repetition rate of 10 MHz. The output power of the laser is lowered to < 1 mW to keep the flux low enough (roughly $150,000$ counts per second on average) to prevent pileup, which is a non-linear effect that distorts the SPAD measurements [1]. The laser emits polarized light that passes through a polarizing beam splitter (Thorlabs PBS251), to a set of 2D scanning mirrors (Thorlabs GVS012). The mirrors are controlled by a multifunction I/O device (NI-DAQ USB-6343) and are used to scan scenes at a spatial resolution of $512{\times}512$ at a rate of 0.1 frames per second. The laser shares an optical path through the beam splitter with a single pixel SPAD (Micro Photon Devices PDM series SPAD) with a 50 $\mu$m $\times$ 50 $\mu$m active pixel area. Photons detected by the SPAD are correlated with a sync signal from the laser using a time-correlated single photon counter (TCSPC) to measure the photon arrival timestamps.

We place the scanned scenes on a rotation stage (Parker Motion/Parker 6K4 Compumotor) in front of the scanning single-photon lidar, allowing us to capture different viewpoints by rotating the scene.

Each scan consists of 20 minutes of total exposure time, but we save out all collected photon timestamps individually so that any desired exposure time can be emulated by accumulating photons over the desired time window in post processing (i.e., for future applications of the dataset). We set the bin width of the photon count histograms to 8 ps and the number of bins is 1500. The entire data acquisition is controlled using custom-developed MATLAB software on a desktop PC, and we capture six scenes with varying geometry, texture, and material properties. For each scene, we capture views in 18 degree increments of the rotation stage, resulting in a 360 degree capture. We set aside 8 views for training, and 10 separate views sampled in 36 degree increments comprise the test split. Prior to input into the network for training, we normalize the measurement values by the maximum photon count observed across all views.

**Ambient illumination.**    Our hardware setup is relatively robust to ambient illumination because we place a laser-line spectral filter (Thorlabs FL532-10) in front of the SPAD, which attenuates ambient illumination by a factor of 10,000. Under indoor lighting, we observe roughly 300-3,000 photon counts per second, depending on the albedo of the target, which is < 2% of the detected laser photons (150,000 counts per second). However, operation in much brighter environments (e.g., outdoors under direct sunlight) would result in non-negligible background counts. In that case, it is possible to use

37th Conference on Neural Information Processing Systems (NeurIPS 2023).

other laser wavelengths (e.g., 1550 nm ), where sunlight is heavily attenuated because of absorption by the atmosphere.

## 2 Calibration of the Hardware Prototype

Here we describe the method used to find the extrinsics and intrinsics defining the captured dataset. An overview of the method can be found in the Algorithm 1.

**Intrinsics**. We calibrate the camera intrinsics of the system using the raxel model [2], which maps each pixel to a 3D ray direction. To find the ray directions, we place a checkerboard on a translation stage and use the lidar system to capture two images of this checkerboard before and after translating 17 cm by in the direction of the surface normal of the board (this was the maximum distance permitted by the translation stage and our optical table layout).

Checkerboard corners are detected in the two images using OpenCV's *findChessboardCorners* [3] with subpixel refinement. In order to improve robustness to distortion, we further refine the corner detection by fitting a second-order polynomial to corners along vertical lines (which we found to be a good fit to model the observed distortion), and we use a first-order fit to horizontal lines, which we found to fit the data well. We set the corner positions to the intersections of the set of fitted lines.

The pixel coordinates of the detected checkerboard corners are then used to define a 3D coordinate system. We set the origin of the coordinate system to the upper-left corner of the nearest checkerboard, and coordinates on the far checkerboard are calculated based on the known translation in $z$, with $x$ and $y$ coordinates given using the size of the checkerboards (4.2 mm). We associate ray directions to each pixel by finding the points of intersection of the ray with the two checkerboards. Specifically, we linearly interpolate the mapping from pixel values to 3D coordinates at each checkerboard corner to retrieve a dense mapping from pixels to 3D coordinates on each board. Then, for a given pixel the ray direction is given by a simple subtraction of the 3D intersection points on each board.

**Extrinsics**. Our captured dataset consists of photon count histograms from six different scenes, each with 20 different views. We use a rotation stage to move the scenes one full revolution in 18 degree increments. The resulting views are equivalent to capturing a stationary scene with cameras that are rotated about the center of rotation of the stage. Since all scenes are captured identically in this fashion, we determine the camera extrinsics once for all scenes.

We also calibrate for an additional offset parameter to finetune the time-of-flight-delays recorded by the lidar system. This accounts for the unknown time delay between the time that the time-correlated single-photon counter receives the sync signal from the laser and the time that the laser pulse reaches the center of projection of the scanning mirrors. In other words, this offset accounts for when "time zero" should occur in the photon count histograms; we roughly calibrate for this value by placing a target directly in front of the galvo mirrors, and we fine tune this offset via optimization as detailed below.

Given that each view is captured using a controlled rotation, the extrinsics can be determined by identifying the 3D axis of rotation and 3D center of rotation. To estimate these parameters, we use a two step procedure. First, we capture lidar scans of a checkerboard placed on the rotation stage and rotated to 6 different positions in 9 degree increments. We convert the lidar scans to a point cloud using the raxel model and lidar time of flight, and then a coarse solution is obtained by fitting planes to the point clouds and finding the center and axis of rotation that align the plane normals. Second, we detect corners of the checkerboards and find the corresponding 3D points. Then, we optimize for the center of rotation, axis of rotation, and the 1D offset parameter that best align the 3D checkerboard corners.

Specifically, we implement a routine in PyTorch [4] to align the 3D checkerboard corners using the Rodrigues formula given below.

$$\mathbf{v}' = (\mathbf{v} - \mathbf{c}) \cos \theta + (\mathbf{a} \cdot (\mathbf{v} - \mathbf{c}))(1 - \cos \theta) \, \mathbf{a} + (\mathbf{a} \times (\mathbf{v} - \mathbf{c})) \sin \theta + \mathbf{c}. \qquad \text{(S1)}$$

Here, $\mathbf{v}$ is the 3D point to be rotated, $\mathbf{c}$ is the center of rotation, $\mathbf{a}$ is the axis of rotation, and $\mathbf{v}'$ is the rotated point. We minimize the objective function given as

$$\mathcal{L}_{\text{align}} = \sum_p \sum_{[i,j] \in \mathcal{C}} \|\mathbf{v}'^{(i)}_p - \mathbf{v}'^{(j)}_p\|_2^2, \qquad \text{(S2)}$$

where $p$ indexes the checkerboard corner points and $[i, j]$ index all pairs of checkerboards. Thus we penalize the distance between corresponding points between all checkerboards. The optimization is performed using LBFGS [5]. After optimization, we use the resulting center point and rotation matrices (i.e., by rotating around **a** by increments of 18 degrees) to define the camera extrinsics for each view.

---

**Algorithm 1:** Calibration overview.

**Intrinsics Calibration**

**Data:** Two scans of a checkerboard before and after translation.

1. Perform sub-pixel detection of checkerboard corners using OpenCV and polynomial line fitting.

2. Use the detected checkerboard corners to initialize a 3D coordinate system.

3. Compute per-pixel 3D coordinates for each checkerboard scan by interpolating the mapping from corner pixel coordinates to 3D coordinates.

4. Compute the ray directions by subtracting the per-pixel 3D coordinates obtained for the checkerboard scan before and after translation.

**Extrinsics Calibration**

**Data:** Six scans of a checkerboard rotated using the rotation stage.

1. Convert lidar scans to point clouds using the time of flight and intrinsics.

2. Fit planes and determine initial center and axis of rotation to align the surface normals.

3. Detect 3D points corresponding to checkerboard corners.

4. Refine the initial center and axis of rotation via Equations S1 and S2.

---

# 3   Additional Implementation Details

**Network Architecture**. In this section we describe the main network architecture. We extend the NerfAcc [6] framework and the provided implementation of Instant-NGP (INGP) [7]. All network hyperparameters are shared between our proposed method and the baselines unless otherwise stated. We set the number of hash feature grids for INGP to 16 and set the feature size to 2. The resolution of the coarsest grid is set to 16 and each subsequent grid has 2 times finer resolution. The base MLP has width 64 with 1 hidden layer to map to density and a latent vector. Another MLP with 2 hidden layers and 64 hidden units maps from the latent vector and viewing direction to the view-dependent radiance.

We use the occupancy grid from NerfAcc with a resolution of $128^3$. The occupancy grid is employed to remove samples along the ray based on their density for the sake of efficiency. The grid is binarized using a occupancy value of $10^{-3}$.

For the captured data, we find that setting the binarization threshold to $10^{-5}$ for the first 3,000 iterations before reverting back to the standard $10^{-3}$ value helps to avoid an overly aggressive removal of density that erodes the surface of reconstructed objects. In addition, we incorporate pruning based on transmittance, removing any samples that register a transmittance of 0 to speed up rendering. Finally, all rays are rendered by sampling 4,096 points along each ray, and these points are pruned according to their occupancy and transmittance values, as previously discussed.

We set the bounding box used in INGP as follows for the simulated and captured data. For the simulated data the bounding box extents are set to -1.5 to 1.5 across all dimensions and methods. For captured data, we use a -0.4 to 0.4 bounding box for Transient NeRF; we slightly shrink the bounding box for baselines run on captured data to -0.3 to 0.3, which we find helps remove some spurious regions of density and improves the baseline results.

**Rendering Equation** Given samples $t_1, t_2, \cdots t_N$ along a camera ray, we render a histogram bin $\tau[n]$ for the ray as

$$\tau[n] = \sum_{i \mid \frac{t_i + t_{i+1}}{2} \in \mathcal{T}_n} T_i^2 \left(1 - \exp\left(-\sigma_i \delta_i\right)\right) \frac{\mathbf{c}_i}{\left(\frac{1}{2}(t_i + t_{i+1})\right)^2}, \text{ where } T_i = \exp\left(-\sum_{j=1}^{i-1} \sigma_j \delta_j\right), \tag{S3}$$

where $\sigma_i, \mathbf{c}_i$ are the density and radiance outputs of the network, $\delta_i = t_{i+1} - t_i$ is the distance between consecutive samples and $\mathcal{T}_n$ is the resolution of the histogram bin, in units of distance. The sum is taken over all consecutive samples whose midpoints fall within $\mathcal{T}_n$.

**Optimization settings and run time**. The time required for training is strongly influenced by the number of samples used for the spatial filter, as discussed in the main text. Specifically, each image pixel is rendered by computing a weighted integral over the radiance for a particular region of space. We compute this integral during training by stochastically sampling one or more ray directions per pixel with probability determined by the weighting and support of the spatial filter [8]. For the simulated dataset, we initially sample a single ray for the first 2000 iterations, then subsequently double this number every 2000 iterations until we reach a maximum of 30 rays sampled per pixel. For the captured dataset we use a single ray sample per pixel per iteration; we find that increasing the number ray samples results in longer training times without significantly improved performance.

Training takes roughly eight hours to converge for the simulated dataset (250K iterations) and two hours to converge for captured data (150K steps).

**Depth calculation**. We follow the NeRF convention in calculating depth for all baseline methods as

$$d = \sum_i \sigma(t_i) T(t_i) \frac{t_i + t_{i+1}}{2}, \tag{S4}$$

where $t_i$ is the distance from the ray origin to a sample along the ray, $T_i$ is the transmittance, and $\sigma$ is the density. Intuitively, this equation calculates the expected ray termination distance.

Since our method incorporates a spatial filter to more accurately model the footprint of the illumination spot, a single ray can result in measurements from multiple surfaces (e.g., if a pixel integrates over a depth discontinuity). To avoid multiple depths in the measurement from skewing the estimate of the expected ray termination distance, we use the following equation to compute depth.

$$d = \arg\max_{t_i} \sigma(t_i) T(t_i). \tag{S5}$$

Thus, we find the depth at which the maximum probability of ray termination occurs.

We note that in the conventional NeRF depth rendering formula of Equation S4, regions of low density become less visible in most visualizations because their depth tends towards zero (i.e., it is weighted by the density, which is close to zero, but typically not uniformly zero for empty regions). However, in Equation S5, no such weighting exists, resulting in visualizations that appear noisier because depths for regions with low (but non-zero) density are not automatically suppressed. Thus for visualization of the depth maps we weight the depth as follows.

$$d = \left(\sum_i \sigma(t_i) T(t_i)\right) \arg\max_{t_i} \sigma(t_i) T(t_i). \tag{S6}$$

We use this equation to visualize depth maps, as we find they are more comparable to those rendered with the conventional NeRF formulation (though the density and transmittance weighting results in them being slightly less accurate).

## 3.1 Baseline Implementation Details

Four different baselines are implemented to evaluate our proposed method. To isolate the impact of different loss functions and speed up the training, we implemented the specific loss functions from each of the following methods in the framework of NerfAcc with the Instant-NGP backbone.

For all baselines, we generate the ground truth intensity images by first integrating the transients over the time dimension and normalization by a scale factor to shift the image values to lie close to within

[0, 1]. We apply gamma correction to tonemap the resulting high dynamic range intensity images prior to training.

**Instant-NGP [7].** The Instant-NGP model is trained with photometric loss defined as the total squared error between the rendered intensities and the pixel colors from the input images:

$$\mathcal{L}_{\text{photo}} = \sum_{\mathbf{r} \in \mathcal{R}} \|\widetilde{\mathbf{C}}(\mathbf{r}) - \mathbf{C}(\mathbf{r})\|_2^2, \tag{S7}$$

where $\widetilde{\mathbf{C}}$ and $\mathbf{C}$ denote the ground-truth pixel color and the predicted value, respectively. $\mathcal{R}$ specifies the set of *active* rays that have non-zero opacity values; this set of rays is updated as training progresses to accelerate optimization by pruning rays that do not contribute to the rendering [6].

**Depth-Supervised NeRF [9].** Following the depth supervision idea proposed in [9], we train a new model that incorporates depth error loss in addition to the photometric loss above:

$$\mathcal{L}_{\text{depth}} = \frac{1}{|\mathcal{R}'|} \sum_{\mathbf{r} \in \mathcal{R}'} \left(\widetilde{d}(\mathbf{r}) - d(\mathbf{r})\right)^2. \tag{S8}$$

Here, $\widetilde{d}(\mathbf{r})$ denotes the lidar depth for ray $\mathbf{r}$ obtained from the captured transients using a log-matched filter [10], $d(\mathbf{r})$ is the predicted depth value computed from Equation S4, and $\mathcal{R}'$ specifies the set of rays that intersect with the object.

The final training loss is defined as $\mathcal{L} = \mathcal{L}_{\text{photo}} + \lambda_{\text{depth}}\mathcal{L}_{\text{depth}}$. In the simulated and captured results, $\lambda_{\text{depth}}$ is set to $0.005$ and $0.0075$, respectively.

**Urban NeRF [11].** We further incorporated the *line-of-sight lidar loss* $\mathcal{L}_{\text{sight}}$, proposed in [11], to encourage the densities to be concentrated near the lidar points. This loss comprises two terms. The first term penalizes any density between the ray origin and the lidar point:

$$\mathcal{L}_{\text{empty}} = \frac{1}{|\mathcal{R}'|} \sum_{\mathbf{r} \in \mathcal{R}'} \Big[ \int_{t_n}^{\widetilde{d}-\epsilon} w(t)^2 dt \Big]. \tag{S9}$$

In this equation, $\mathcal{R}'$ is the set of rays that intersect with the object; $w(t)$ is the volume rendering integration weights defined as $\sigma(t)T(t)$; $t_n$ denotes the near bound of the ray; and $\epsilon$ specifies a neighbourhood around the surface point.

The second term, on the other hand, encourages the model to increase the densities in the bounded region around the surface point:

$$\mathcal{L}_{\text{near}} = \frac{1}{|\mathcal{R}'|} \sum_{\mathbf{r} \in \mathcal{R}'} \Big[ \int_{\widetilde{d}-\epsilon}^{\widetilde{d}+\epsilon} (w(t) - \mathcal{K}_\epsilon(t - \widetilde{d}))^2 dt \Big], \tag{S10}$$

where $\mathcal{K}_\epsilon$ is a truncated Gaussian defined as $\mathcal{N}(0, (\epsilon/3)^2)$ [12].

The final training loss for this model is defined as:

$$\mathcal{L} = \mathcal{L}_{\text{photo}} + \lambda_{\text{depth}}\mathcal{L}_{\text{depth}} + \lambda_{\text{sight}}\mathcal{L}_{\text{sight}}, \tag{S11}$$

where $\mathcal{L}_{\text{sight}} = \mathcal{L}_{\text{empty}} + \mathcal{L}_{\text{near}}$. The parameters $(\lambda_{\text{depth}}, \lambda_{\text{sight}})$ are set to $(0.0001, 0.005)$ for both the simulations and captured experiments.

Following the instructions from the original paper, we applied exponential decay to the value of $\epsilon$ throughout the optimization, which encourages the density of the radiance field to fall within a progressively smaller support, improving convergence. We initialize $\epsilon$ to a value $\epsilon_{\max}$, and every 7000 steps we multiply by a factor of $0.8$, until it decays to $\epsilon_{\min}$. Parameters $(\epsilon_{\max}, \epsilon_{\min})$ are set to $(1.5, 0.025)$ and $(4.0, 0.05)$ for the captured and simulated results, respectively.

**Urban NeRF with Masking.** The depth-related loss terms $\mathcal{L}_{\text{depth}}$ and $\mathcal{L}_{\text{sight}}$ used in the previous two models are evaluated only for the rays that intersect with the object; however, we find that the included regularization terms do not prevent spurious regions of density that appear for pixels that fall outside the support of the object on the image plane.

We attempt to improve the baseline results further by using modified versions of $\mathcal{L}_{\text{photo}}$ and $\mathcal{L}_{\text{sight}}$ which sum over active rays $\mathcal{R}$. Specifically we use an oracle object mask (i.e., a ground truth mask

segmenting the object in each training view) and set the ground truth depth of all background pixels to zero. This loss encourages the density to be uniformly zero along background rays and helps to remove spurious clouds of density that tend to materialize in the other baseline results during training. We note that this masking loss is directly analogous to the *sky modeling* loss proposed in Urban-NeRF [11], which penalizes density along rays that intersect with the sky as determined using a semantic segmentation network.

For both the simulated and captured results, we hand-annotate the oracle object masks by carefully thresholding the intensity images and applying morphology operations to fill in holes in the mask.

# 4 Supplemental Results

## 4.1 Dataset

**Simulated dataset.** To create the simulated multiview lidar dataset, we modify the non-line-of-sight Mitsuba 2 rendering codes of Royo et al. [13]. While the original codebase enables one to simulate the effect of illuminating a single point in the scene with a laser and imaging other points, our approach requires rendering a frame where each pixel images an area of the scene that is illuminated with a coaxial, collimated light source. To this end, introduce a new light source as well as additional rendering options such that each pixel is rendered independently with its own coaxial light source. We also incorporate a Gaussian reconstruction filter [8] along the time dimension to avoid aliasing or stair-stepping artifacts in regions with fine variations in depth.

We choose the variance of the Gaussian temporal filter and spatial filters to produce transients that measure a laser pulse with similar temporal and spatial support as the captured results. Specifically, we set the variance of the temporal and spatial filters to $3$ and $0.15$ respectively. We simulate photon count histograms 1200 bins and each bin has a width of approximately $\sim 30ps$.

We subsample 8 training viewpoints from the simulated lidar measurements rendered at 36 degree increments along a circle centered at the origin. Specifically, we sample 2 views separated by 180 degrees; 3 views separated by 90 and 180 degrees (i.e., a superset of the 2 views), and 5 views uniformly separated by 72 degrees. Note that the 5 views are not a superset of the viewpoints used when training on 2 or 3 lidar measurements. For testing, we use 6 viewpoints from the NeRF Blender test set that surround the object.

**Captured dataset.** The captured dataset consists of 20 multiview single-photon lidar scans of six scenes. The scenes consist of everyday objects and figurines (see Fig. S1). We record photon timestamps at 4 ps resolution for each scene, with measurements of each view being captured during an exposure period of 20 minutes. While we use all photon timestamps in the photon count histograms used for the proposed method, access to the photon timestamps also allows synthesizing histograms with arbitrary exposure time, which will make the dataset useful for a wide array of follow-on work.

We use raw photon count histograms with 4096 bins and bin widths of $4$ ps. To decrease the memory required for training, we crop and downsample the histograms to 1500 bins with 8 ps resolution.

Measurements are captured in the low-flux regime to avoid non-linear distortions due to pile-up. A detailed breakdown of the photon counts and acquisition parameters for each captured scene is provided in Table S1.

Table S1: Photon counts per scene for the captured dataset.

| scene name | spatial resolution | histogram bins | exposure time/view | avg. counts/view | avg. counts/sec |
|---|---|---|---|---|---|
| *baskets* | $512 \times 512$ | 1500 | 20 min | $9.04 \times 10^7$ | $7.53 \times 10^4$ |
| *boots* | | | | $7.39 \times 10^7$ | $6.16 \times 10^4$ |
| *carving* | | | | $5.60 \times 10^7$ | $4.66 \times 10^4$ |
| *chef* | | | | $2.56 \times 10^8$ | $2.13 \times 10^5$ |
| *cinema* | | | | $1.51 \times 10^8$ | $1.26 \times 10^5$ |
| *food* | | | | $1.44 \times 10^8$ | $1.20 \times 10^5$ |

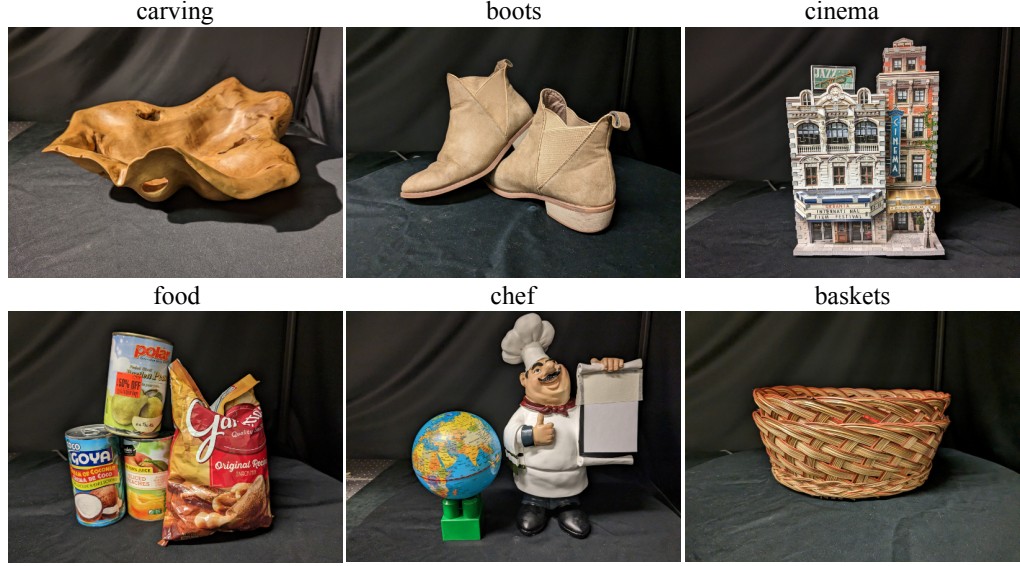

Figure S1: The captured dataset. The captured datset consists of 20 multiview single-photon lidar scans of six scenes.

## 4.2 Depth Evaluation

DS-NeRF and Urban NeRF calculate depth differentiably using the integral along the ray (i.e., the expected ray termination distance) and incorporate this in their loss functions. Since the baseline methods are supervised with this depth estimate, we opted to use the same approach for evaluation.

For completeness, we provide an updated version of the L1 depth in Table S2 calculated for all methods using the maximum ray termination probability as described in Equation S5. Baseline performance using this method of depth estimation is indeed improved for most of the methods, though Transient NeRF still performs best.

Table S2: Depth evaluation using maximum probability of ray termination for simulated and captured scenes.

| Method | Simulated ↓ | | | Captured ↓ | | |
|---|---|---|---|---|---|---|
| | 2 views | 3 views | 5 views | 2 views | 3 views | 5 views |
| Instant NGP [30] | 0.151 | 0.202* | 0.184* | 0.021 | 0.021 | 0.024 |
| DS-NeRF [26] | 0.102 | 0.124* | 0.136* | 0.024 | 0.026 | 0.033 |
| Urban NeRF [4] | 0.128 | 0.124 | 0.100 | 0.017 | 0.016* | 0.013 |
| Urban NeRF w/mask† | 0.039 | 0.038 | 0.020 | 0.014 | 0.006 | 0.004 |
| Proposed | 0.015 | 0.011 | 0.013 | 0.006 | 0.006 | 0.010 |

* Using maximum probability of ray termination is worse than the expected termination distance (Equation S4).
† Requires ground truth segmentation mask.

## 4.3 Intermediate Rendering Results

In Fig. S2, we show intermediate results from the rendering pipeline: the density from the network and the raw radiance output. We plot all these quantities versus the histogram bin number, which we find by simply discretizing the distances of the samples along the rays and allocating the quantities to the appropriate bin.

The rendered densities are very dissimilar from the rendered transients, partly due to the transients being effected by the transmittance, but also because these values are visualized before applying the temporal filter.

Note these plots show renders for a single ray, and we are not integrating over multiple rays to account for the pixel footprint. Also, we plot the normalized density and radiance along the ray since these quantities vary significantly across different pixels in the image.

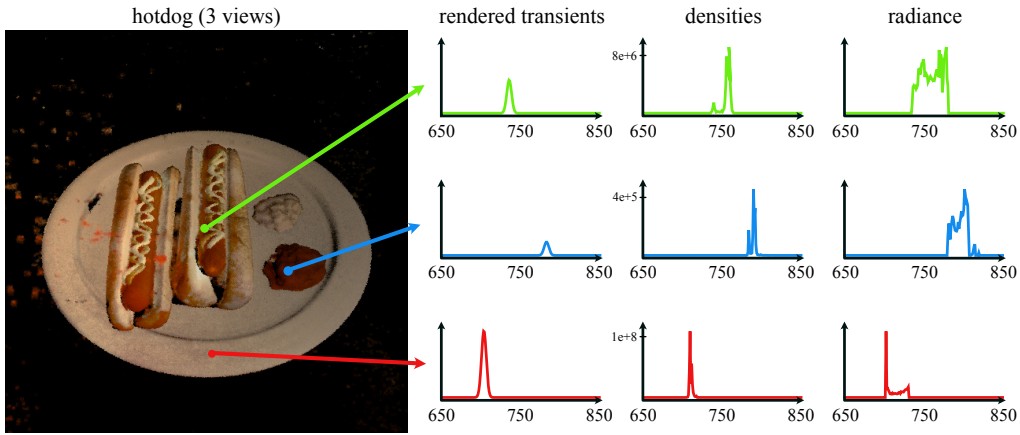

Figure S2: Rendered transients, densities, and radiance plotted versus bin number for rays represented in the rendered image of the *hotdog* scene trained on three views. We normalize the densities and radiance values for visualiation and plot the unnormalized transients.

## 4.4 Ablation Studies

**Space carving, temporal filter and normals.** In Table S3 we show ablation studies of our method calculated on the captured *cinema* scene. Specifically, we include quantitative results of our method without the space carving loss (w/o SC), without accounting for the laser profile (w/o TF) and the proposed method while accounting for normals (w/ normals).

The space carving loss appears to especially improve performance in the five view case. It helps eliminate spurious clouds of density, which improves reconstruction quality for held-out test views.

We also note the importance of explicitly accounting for the laser pulse width and system jitter. This is especially noticeable for the two view results, where the depth becomes highly skewed without this component. We attribute this effect to a thickening of the density representing the object surface; the optimization appears to converge to this solution to explain the temporal spread of the returning light captured in the photon count histograms. Properly accounting for the laser pulse and system jitter essentially deconvolves the temporal response of the lidar system, resulting in thin sheets of density that accurately localize the object surface.

We also ablate using the normals by adding cosine factors to the rendering equation (Eq. 3) for the Cinema scene. Estimating the normals requires a noisy finite-difference operation on the predicted density values, which we found results in speckle-like artifacts in the novel views. Instead, we model the effects of incidence angle by conditioning the neural representation on view direction.

Table S3: Ablation studies on the proposed method for the captured cinema scene. We present results while omitting any temporal filtering (w/o TF), omitting the space carving loss (w/o SC) and with modelling normals using cosine factors (w/ normals).

| Method | PSNR (dB) ↑ | | | LPIPS ↓ | | | SSIM ↑ | | | L1 (depth) ↓ | | |
|---|---|---|---|---|---|---|---|---|---|---|---|---|
| | 2 views | 3 views | 5 views | 2 views | 3 views | 5 views | 2 views | 3 views | 5 views | 2 views | 3 views | 5 views |
| Proposed w/o TF | 16.55 | 20.85 | 20.02 | 0.346 | 0.225 | 0.209 | 0.589 | 0.837 | 0.823 | 0.022 | 0.007 | 0.009 |
| Proposed w/o SC | 20.63 | 21.09 | 20.12 | 0.207 | 0.200 | 0.217 | 0.855 | 0.855 | 0.840 | 0.007 | 0.009 | 0.020 |
| Proposed w/ normals | 20.88 | 20.41 | 24.62 | 0.332 | 0.268 | 0.216 | 0.825 | 0.815 | 0.867 | 0.006 | 0.007 | 0.007 |
| Proposed | 21.61 | 21.66 | 25.12 | 0.281 | 0.245 | 0.178 | 0.850 | 0.812 | 0.879 | 0.006 | 0.006 | 0.007 |

**Pixel footprint.** We ablate modelling the pixel footprint (see Table S4). When the laser spot and sensor footprint pass over a depth discontinuity, we observe two peaks in the resulting photon count histogram (corresponding to the two depths across the discontinuity). Assuming an ideal spot (i.e., using a single ray) completely fails to model this effect. Training and rendering the *lego* scene (which has many depth discontinuities) with the ideal spot model results in much worse performance across novel views.

Table S4: Simulated results comparing the proposed approach on the *Lego* scene with and without modeling the laser spatial footprint (w/o SF).

| Method | PSNR (dB) ↑ | | | LPIPS ↓ | | | L1 (depth) ↓ | | |
|---|---|---|---|---|---|---|---|---|---|
| | 2 views | 3 views | 5 views | 2 views | 3 views | 5 views | 2 views | 3 views | 5 views |
| Proposed w/o SF | 20.01 | 23.39 | 25.08 | 0.195 | 0.147 | 0.170 | 0.119 | 0.065 | 0.041 |
| Proposed | 20.64 | 23.63 | 25.18 | 0.190 | 0.161 | 0.192 | 0.023 | 0.011 | 0.008 |

**HDR-informed loss function.** We provide an ablation study of the HDR-informed loss function in Table S5 on the *chef* and *food* scenes. Incorporating this loss provides some improvement by preventing very bright regions from dominating the loss. Moreover, in the case of 5 input views, performance without the HDR-informed loss drops due to specular highlights appearing in one view, but not an overlapping nearby training view (e.g., on the globe in the *chef* scene or the bag of chips in the *food* scene). Without the HDR-informed loss, the network has difficulty modeling the large variation in radiance between views, and so the optimization produces spurious patches of density to model this view-dependent effect (despite regularization with the space carving loss).

Table S5: Ablation study of the HDR-informed loss function for the captured *Chef* and *Food* scenes.

| Scene/Method | PSNR (dB) ↑ | | | LPIPS ↓ | | | SSIM ↑ | | | L1 (depth) ↓ | | |
|---|---|---|---|---|---|---|---|---|---|---|---|---|
| | 2 views | 3 views | 5 views | 2 views | 3 views | 5 views | 2 views | 3 views | 5 views | 2 views | 3 views | 5 views |
| *Chef* w/o HDR | 18.19 | 18.09 | 13.54 | 0.360 | 0.324 | 0.345 | 0.767 | 0.773 | 0.643 | 0.004 | 0.005 | 0.025 |
| *Chef* w/ HDR | 19.27 | 18.14 | 19.55 | 0.334 | 0.338 | 0.276 | 0.775 | 0.747 | 0.811 | 0.006 | 0.007 | 0.013 |
| *Food* w/o HDR | 23.08 | 23.98 | 16.93 | 0.305 | 0.184 | 0.263 | 0.827 | 0.881 | 0.722 | 0.007 | 0.006 | 0.032 |
| *Food* w/ HDR | 23.40 | 23.78 | 22.09 | 0.286 | 0.205 | 0.154 | 0.826 | 0.879 | 0.873 | 0.006 | 0.007 | 0.013 |

## 4.5 Simulated Results

The paper focuses on results for sparse views (i.e., 2, 3, and 5 views) since this is the regime where depth supervision improves the most over only using 2D supervision. However in Table S6 we show results of the proposed method and the baselines trained on 10 simulated training views sampled at equal angles around the Lego scene, and our approach still outperforms the baselines when evaluated on the same test views.

Table S6: Simulated results for 2, 3, 5, and 10 training views on the *Lego* scene.

| Method | PSNR (dB) ↑ | | | | LPIPS ↓ | | | | L1 (depth) ↓ | | | |
|---|---|---|---|---|---|---|---|---|---|---|---|---|
| | 2 views | 3 views | 5 views | 10 views | 2 views | 3 views | 5 views | 10 views | 2 views | 3 views | 5 views | 10 views |
| Instant NGP [30] | 16.04 | 17.46 | 16.46 | 21.73 | 0.591 | 0.544 | 0.479 | 0.216 | 0.224 | 0.236 | 0.195 | 0.101 |
| DS-NeRF [26] | 17.66 | 19.45 | 17.21 | 22.83 | 0.500 | 0.476 | 0.482 | 0.302 | 0.136 | 0.127 | 0.193 | 0.068 |
| Urban NeRF [4] | 19.59 | 19.75 | 16.19 | 21.36 | 0.519 | 0.522 | 0.503 | 0.368 | 0.105 | 0.100 | 0.151 | 0.074 |
| Urban NeRF w/mask | 20.48 | 21.50 | 19.48 | 25.02 | 0.471 | 0.442 | 0.421 | 0.317 | 0.040 | 0.049 | 0.057 | 0.019 |
| Proposed | 20.64 | 23.63 | 25.18 | 31.74 | 0.190 | 0.161 | 0.192 | 0.084 | 0.023 | 0.011 | 0.008 | 0.005 |

In Figs. S3, S4, and S5 we show further renders from our method and baseline methods trained on the simulated dataset. The same trends as in the main text persist. Our method qualitatively produces images more faithful to the ground-truth. Our rendered images suffer less from floating artifacts and display finer details, for example in the *lego* scene. Some artifacts in the depth maps (i.e., the "holes" that appear) result from how we visualize depth in low occupancy areas. Depths in these regions can be become biased as described by Equation S6.

We show a breakdown across all simulated scenes of the evaluation metrics (see Table S7, S8, S9, S10). In addition to the metrics reported in the main text (PSNR, LPIPS, L1 depth) we add the structural similarity (SSIM) metric. Again, our method outperforms the baselines in the quantitative metrics. Since the 2, 3, and 5 view results do not include viewpoints that are strict supersets of each other, the performance of some metrics does not always increase for every scene in every case with increasing views (though this is a trend we observe on average).

Table S7: Breakdown of PSNR (dB) across all 5 simulated scenes.

| Scene | Instant NGP [7] ↑ | | | DS-NeRF [9] | | | Urban NeRF [12] | | | Urban NeRF w/Mask [12] | | | Proposed | | |
|---|---|---|---|---|---|---|---|---|---|---|---|---|---|---|---|
| | 2 views | 3 views | 5 views | 2 views | 3 views | 5 views | 2 views | 3 views | 5 views | 2 views | 3 views | 5 views | 2 views | 3 views | 5 views |
| lego | 16.04 | 17.46 | 16.46 | 17.66 | 19.45 | 17.21 | 19.59 | 19.75 | 16.19 | 20.48 | 21.50 | 19.48 | 20.64 | 23.63 | 25.81 |
| chair | 13.40 | 15.42 | 24.95 | 16.51 | 16.72 | 25.49 | 17.53 | 16.14 | 24.28 | 20.39 | 18.69 | 28.23 | 20.75 | 21.99 | 34.48 |
| hotdog | 14.95 | 14.72 | 13.80 | 18.93 | 17.47 | 19.12 | 15.75 | 16.37 | 18.04 | 18.90 | 18.71 | 19.53 | 20.74 | 22.64 | 32.36 |
| bench | 16.85 | 19.33 | 15.58 | 19.87 | 20.34 | 15.69 | 19.12 | 19.52 | 14.56 | 20.64 | 21.55 | 17.57 | 20.20 | 23.06 | 21.57 |
| ficus | 21.87 | 21.40 | 27.50 | 23.44 | 22.75 | 27.85 | 22.32 | 21.88 | 25.91 | 24.12 | 23.58 | 26.87 | 24.57 | 26.10 | 27.70 |
| average | 16.62 | 17.67 | 19.66 | 19.28 | 19.35 | 21.07 | 18.86 | 18.73 | 19.80 | 20.91 | 20.81 | 22.34 | 21.38 | 23.48 | 28.39 |

Table S8: Breakdown of LPIPS metric across all 5 simulated scenes.

| Scene | Instant NGP [7] ↑ | | | DS-NeRF [9] | | | Urban NeRF [12] | | | Urban NeRF w/Mask [12] | | | Proposed | | |
|---|---|---|---|---|---|---|---|---|---|---|---|---|---|---|---|
| | 2 views | 3 views | 5 views | 2 views | 3 views | 5 views | 2 views | 3 views | 5 views | 2 views | 3 views | 5 views | 2 views | 3 views | 5 views |
| lego | 0.591 | 0.544 | 0.479 | 0.500 | 0.476 | 0.482 | 0.519 | 0.522 | 0.503 | 0.471 | 0.442 | 0.421 | 0.190 | 0.161 | 0.192 |
| chair | 0.501 | 0.466 | 0.296 | 0.482 | 0.461 | 0.306 | 0.494 | 0.474 | 0.326 | 0.359 | 0.357 | 0.275 | 0.138 | 0.138 | 0.037 |
| hotdog | 0.583 | 0.615 | 0.523 | 0.456 | 0.470 | 0.390 | 0.580 | 0.553 | 0.433 | 0.533 | 0.466 | 0.395 | 0.242 | 0.241 | 0.118 |
| bench | 0.538 | 0.442 | 0.398 | 0.444 | 0.422 | 0.459 | 0.502 | 0.473 | 0.483 | 0.400 | 0.368 | 0.375 | 0.194 | 0.139 | 0.159 |
| ficus | 0.387 | 0.311 | 0.238 | 0.272 | 0.352 | 0.244 | 0.403 | 0.400 | 0.286 | 0.287 | 0.278 | 0.229 | 0.094 | 0.079 | 0.069 |
| average | 0.520 | 0.476 | 0.387 | 0.431 | 0.436 | 0.376 | 0.500 | 0.484 | 0.406 | 0.410 | 0.382 | 0.339 | 0.172 | 0.151 | 0.115 |

Table S9: Breakdown of SSIM metric across all 5 simulated scenes.

| Scene | Instant NGP [7] ↑ | | | DS-NeRF [9] | | | Urban NeRF [12] | | | Urban NeRF w/Mask [12] | | | Proposed | | |
|---|---|---|---|---|---|---|---|---|---|---|---|---|---|---|---|
| | 2 views | 3 views | 5 views | 2 views | 3 views | 5 views | 2 views | 3 views | 5 views | 2 views | 3 views | 5 views | 2 views | 3 views | 5 views |
| lego | 0.456 | 0.519 | 0.576 | 0.600 | 0.624 | 0.631 | 0.592 | 0.580 | 0.590 | 0.646 | 0.710 | 0.722 | 0.860 | 0.897 | 0.899 |
| chair | 0.545 | 0.632 | 0.767 | 0.653 | 0.659 | 0.760 | 0.582 | 0.624 | 0.766 | 0.774 | 0.756 | 0.897 | 0.901 | 0.899 | 0.977 |
| hotdog | 0.508 | 0.448 | 0.534 | 0.736 | 0.636 | 0.687 | 0.503 | 0.534 | 0.664 | 0.567 | 0.686 | 0.761 | 0.882 | 0.875 | 0.963 |
| bench | 0.537 | 0.591 | 0.588 | 0.703 | 0.664 | 0.618 | 0.554 | 0.615 | 0.580 | 0.717 | 0.769 | 0.747 | 0.856 | 0.884 | 0.870 |
| ficus | 0.682 | 0.819 | 0.851 | 0.831 | 0.789 | 0.873 | 0.734 | 0.711 | 0.844 | 0.819 | 0.840 | 0.913 | 0.925 | 0.934 | 0.942 |
| average | 0.546 | 0.602 | 0.663 | 0.705 | 0.675 | 0.714 | 0.593 | 0.613 | 0.689 | 0.705 | 0.752 | 0.808 | 0.885 | 0.898 | 0.930 |

Table S10: Breakdown of L1 (depth) metric across all 5 simulated scenes.

| Scene | Instant NGP [7] ↑ | | | DS-NeRF [9] | | | Urban NeRF [12] | | | Urban NeRF w/Mask [12] | | | Proposed | | |
|---|---|---|---|---|---|---|---|---|---|---|---|---|---|---|---|
| | 2 views | 3 views | 5 views | 2 views | 3 views | 5 views | 2 views | 3 views | 5 views | 2 views | 3 views | 5 views | 2 views | 3 views | 5 views |
| lego | 0.224 | 0.236 | 0.195 | 0.136 | 0.127 | 0.193 | 0.105 | 0.100 | 0.151 | 0.040 | 0.049 | 0.057 | 0.023 | 0.011 | 0.008 |
| chair | 0.166 | 0.198 | 0.117 | 0.123 | 0.146 | 0.097 | 0.130 | 0.132 | 0.066 | 0.034 | 0.057 | 0.009 | 0.006 | 0.005 | 0.003 |
| hotdog | 0.344 | 0.254 | 0.271 | 0.131 | 0.151 | 0.115 | 0.301 | 0.270 | 0.089 | 0.126 | 0.040 | 0.008 | 0.007 | 0.006 | 0.003 |
| bench | 0.166 | 0.137 | 0.198 | 0.082 | 0.076 | 0.142 | 0.060 | 0.047 | 0.168 | 0.011 | 0.007 | 0.038 | 0.006 | 0.005 | 0.003 |
| ficus | 0.291 | 0.147 | 0.107 | 0.072 | 0.074 | 0.048 | 0.059 | 0.073 | 0.034 | 0.043 | 0.034 | 0.032 | 0.034 | 0.033 | 0.048 |
| average | 0.238 | 0.195 | 0.178 | 0.109 | 0.115 | 0.119 | 0.131 | 0.124 | 0.101 | 0.051 | 0.038 | 0.029 | 0.015 | 0.011 | 0.013 |

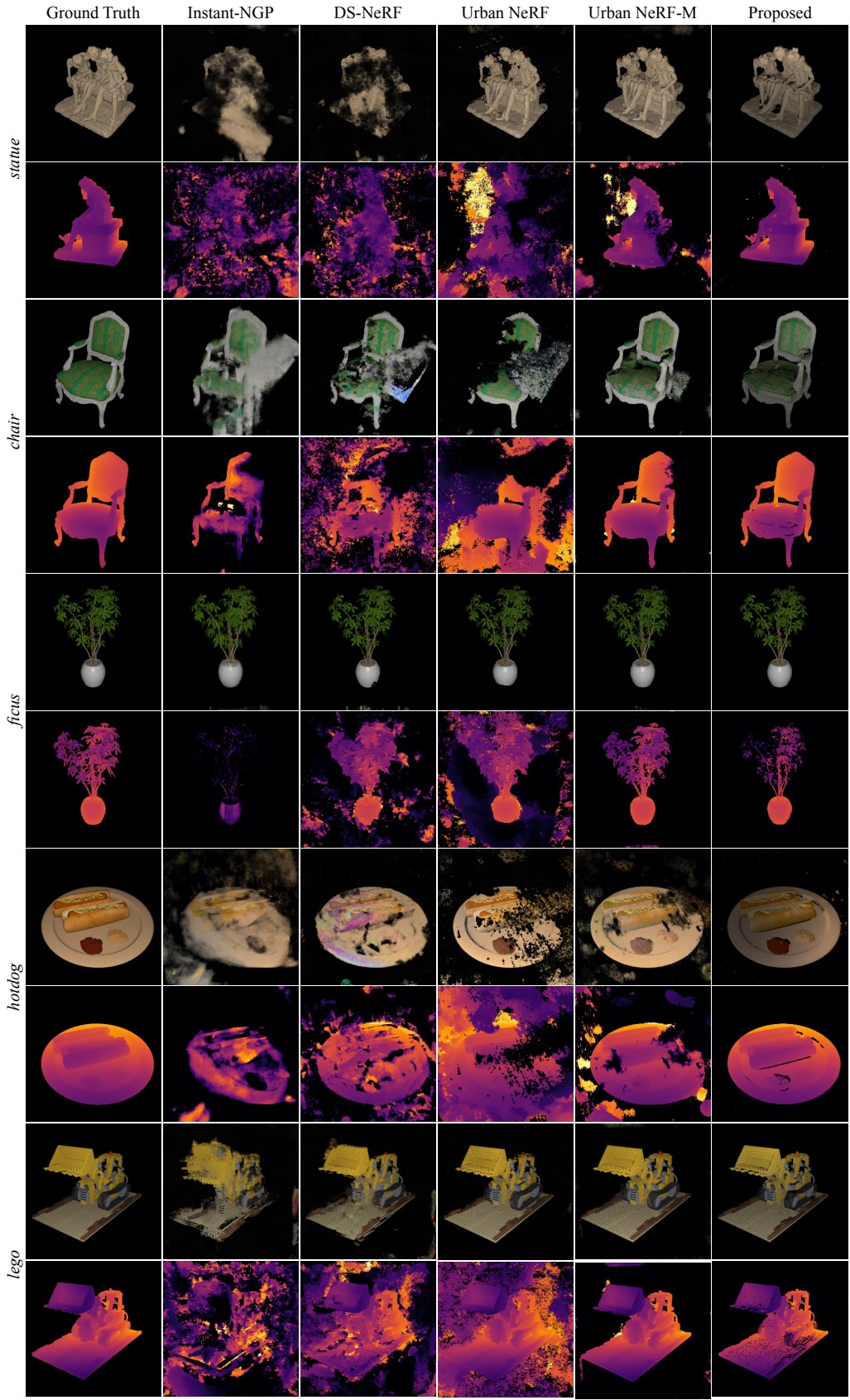

Figure S3: Rendered images and depths on the simulated dataset for 2 views.

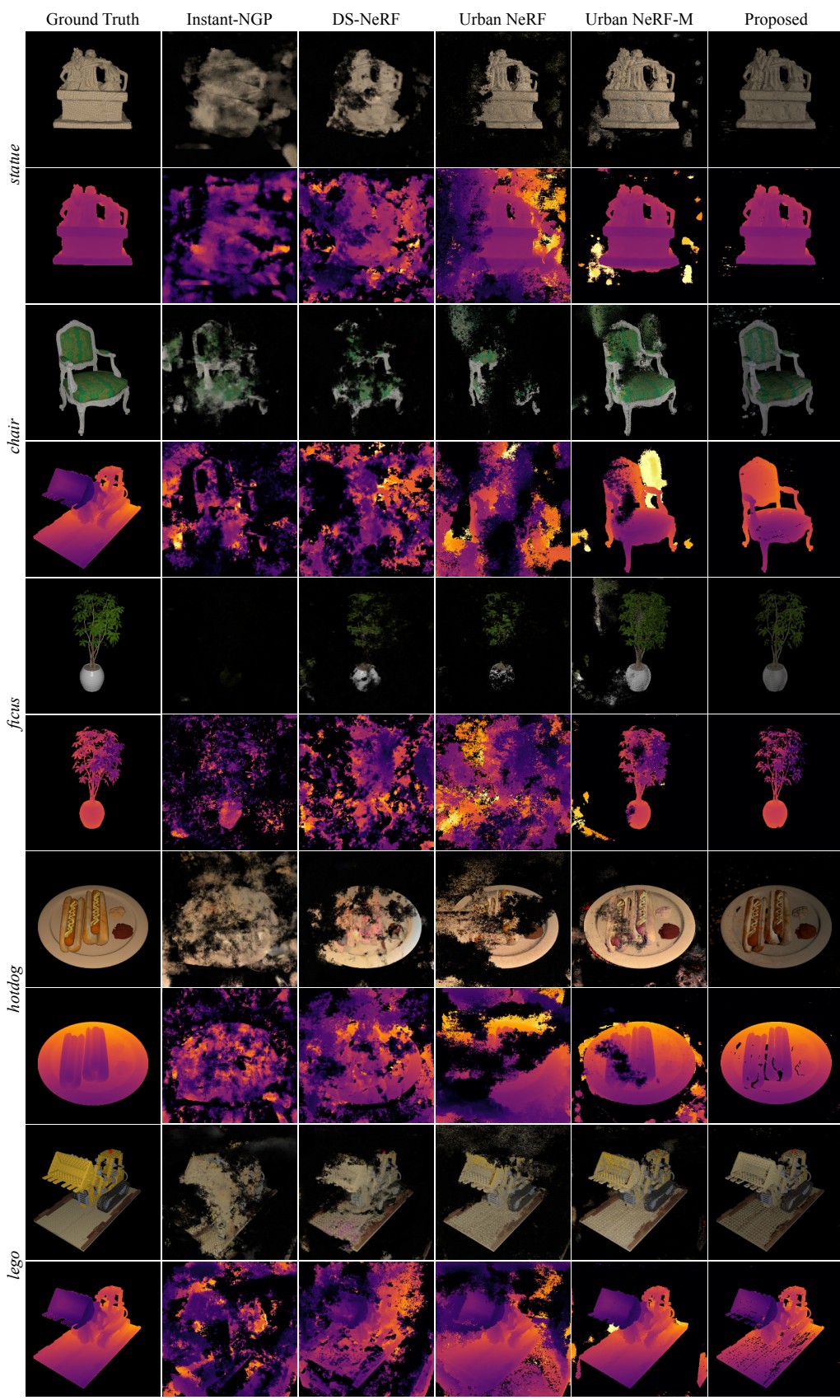

Figure S4: Rendered images and depths on the simulated dataset for 3 views.

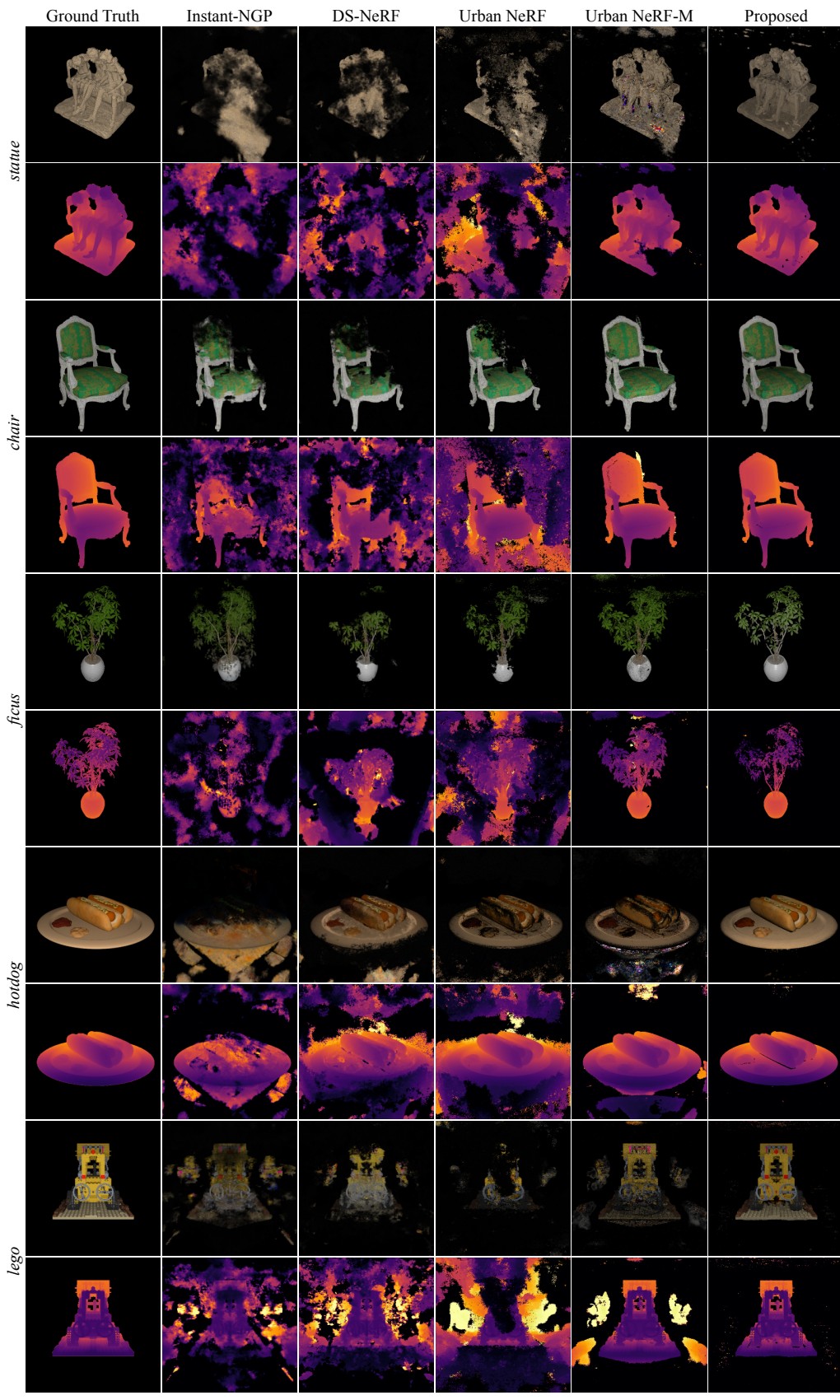

Figure S5: Rendered images and depths on the simulated dataset for 5 views.

## 4.6 Captured Results

In Figs. S6, S7, and S8 we show further renders from our model and the baselines on our captured dataset. The proposed method produces images more faithful to the ground truth.

We note that the quality of the reconstructions is lower than the simulation results. We attribute this to small imperfections in the calibration of the camera extrinsics, which register the multiview lidar scans to an accuracy of approximately 1 mm. Achieving precise, sub-mm alignment of the multiview lidar scans is a highly non-trivial problem and is outside the scope of our current work. The captured results bear out the trends observed in simulation and demonstrate Transient NeRF and novel view synthesis of lidar measurements for the first time in practice.

We show a breakdown across all captured scenes of the evaluation metrics (see Tables S11, S12, S13, and S14). In addition to the metrics reported in the main text (PSNR, LPIPS, L1 depth) we add the SSIM metric. Again our method outperforms the baselines in the quantitative metrics.

Table S11: Breakdown of PSNR (dB) across all 6 captured scenes.

| Scene | Instant NGP [7] ↑ | | | DS-NeRF [9] | | | Urban NeRF [12] | | | Urban NeRF w/Mask [12] | | | Proposed | | |
|---|---|---|---|---|---|---|---|---|---|---|---|---|---|---|---|
| | 2 views | 3 views | 5 views | 2 views | 3 views | 5 views | 2 views | 3 views | 5 views | 2 views | 3 views | 5 views | 2 views | 3 views | 5 views |
| cinema | 14.73 | 15.18 | 15.21 | 13.18 | 13.87 | 12.58 | 17.27 | 15.11 | 17.30 | 14.13 | 17.22 | 17.80 | 21.61 | 21.66 | 25.12 |
| boots | 16.54 | 18.32 | 18.09 | 16.97 | 15.65 | 16.59 | 16.54 | 14.54 | 16.59 | 13.93 | 18.70 | 21.87 | 22.38 | 22.29 | 24.94 |
| baskets | 18.73 | 19.78 | 18.17 | 16.47 | 17.43 | 17.54 | 16.68 | 15.88 | 13.42 | 17.17 | 16.51 | 14.26 | 22.48 | 20.93 | 19.90 |
| carving | 17.96 | 17.30 | 16.70 | 17.41 | 16.36 | 15.44 | 18.71 | 17.97 | 17.36 | 16.07 | 21.41 | 22.28 | 23.52 | 24.20 | 24.70 |
| chef | 13.04 | 11.72 | 13.76 | 12.32 | 12.27 | 11.51 | 13.34 | 13.71 | 13.15 | 14.06 | 15.25 | 16.72 | 19.27 | 18.14 | 19.55 |
| food | 17.64 | 16.83 | 16.45 | 15.68 | 14.70 | 15.48 | 18.86 | 18.27 | 17.77 | 17.37 | 20.45 | 21.72 | 23.40 | 23.78 | 22.09 |
| average | 16.44 | 16.52 | 16.39 | 15.34 | 15.05 | 14.86 | 16.90 | 15.91 | 15.93 | 15.45 | 18.26 | 19.11 | 22.11 | 21.83 | 22.72 |

Table S12: Breakdown of LPIPS across all 6 captured scenes.

| Scene | Instant NGP [7] ↑ | | | DS-NeRF [9] | | | Urban NeRF [12] | | | Urban NeRF w/Mask [12] | | | Proposed | | |
|---|---|---|---|---|---|---|---|---|---|---|---|---|---|---|---|
| | 2 views | 3 views | 5 views | 2 views | 3 views | 5 views | 2 views | 3 views | 5 views | 2 views | 3 views | 5 views | 2 views | 3 views | 5 views |
| cinema | 0.445 | 0.374 | 0.314 | 0.396 | 0.364 | 0.429 | 0.369 | 0.350 | 0.273 | 0.457 | 0.295 | 0.244 | 0.281 | 0.245 | 0.178 |
| boots | 0.251 | 0.253 | 0.181 | 0.193 | 0.225 | 0.192 | 0.386 | 0.332 | 0.163 | 0.525 | 0.245 | 0.135 | 0.221 | 0.182 | 0.155 |
| baskets | 0.399 | 0.298 | 0.252 | 0.256 | 0.244 | 0.268 | 0.444 | 0.311 | 0.220 | 0.433 | 0.300 | 0.195 | 0.269 | 0.165 | 0.164 |
| carving | 0.174 | 0.167 | 0.183 | 0.168 | 0.186 | 0.225 | 0.357 | 0.217 | 0.146 | 0.436 | 0.170 | 0.125 | 0.232 | 0.138 | 0.103 |
| chef | 0.580 | 0.443 | 0.401 | 0.498 | 0.434 | 0.467 | 0.513 | 0.468 | 0.362 | 0.473 | 0.373 | 0.270 | 0.334 | 0.338 | 0.276 |
| food | 0.299 | 0.310 | 0.313 | 0.354 | 0.417 | 0.369 | 0.351 | 0.290 | 0.221 | 0.425 | 0.232 | 0.174 | 0.286 | 0.205 | 0.154 |
| average | 0.358 | 0.307 | 0.274 | 0.311 | 0.312 | 0.325 | 0.403 | 0.328 | 0.231 | 0.458 | 0.269 | 0.191 | 0.271 | 0.212 | 0.172 |

Table S13: Breakdown of SSIM across all 6 captured scenes.

| Scene | Instant NGP [7] ↑ | | | DS-NeRF [9] | | | Urban NeRF [12] | | | Urban NeRF w/Mask [12] | | | Proposed | | |
|---|---|---|---|---|---|---|---|---|---|---|---|---|---|---|---|
| | 2 views | 3 views | 5 views | 2 views | 3 views | 5 views | 2 views | 3 views | 5 views | 2 views | 3 views | 5 views | 2 views | 3 views | 5 views |
| cinema | 0.545 | 0.628 | 0.717 | 0.620 | 0.666 | 0.591 | 0.602 | 0.673 | 0.768 | 0.509 | 0.728 | 0.807 | 0.850 | 0.812 | 0.879 |
| boots | 0.804 | 0.810 | 0.866 | 0.853 | 0.822 | 0.851 | 0.587 | 0.687 | 0.871 | 0.414 | 0.777 | 0.894 | 0.912 | 0.909 | 0.914 |
| baskets | 0.555 | 0.715 | 0.749 | 0.737 | 0.750 | 0.738 | 0.499 | 0.689 | 0.751 | 0.517 | 0.691 | 0.771 | 0.846 | 0.850 | 0.826 |
| carving | 0.852 | 0.853 | 0.843 | 0.858 | 0.841 | 0.804 | 0.620 | 0.827 | 0.877 | 0.526 | 0.854 | 0.906 | 0.913 | 0.929 | 0.929 |
| chef | 0.414 | 0.586 | 0.647 | 0.544 | 0.608 | 0.574 | 0.450 | 0.559 | 0.686 | 0.488 | 0.651 | 0.780 | 0.775 | 0.747 | 0.811 |
| food | 0.731 | 0.720 | 0.733 | 0.677 | 0.586 | 0.671 | 0.654 | 0.737 | 0.808 | 0.528 | 0.775 | 0.855 | 0.826 | 0.879 | 0.873 |
| average | 0.650 | 0.719 | 0.759 | 0.715 | 0.712 | 0.705 | 0.569 | 0.695 | 0.793 | 0.497 | 0.746 | 0.836 | 0.854 | 0.854 | 0.872 |

Table S14: Breakdown of L1 (depth) metric across all 6 captured scenes.

| Scene | Instant NGP [7] ↑ | | | DS-NeRF [9] | | | Urban NeRF [12] | | | Urban NeRF w/Mask [12] | | | Proposed | | |
|---|---|---|---|---|---|---|---|---|---|---|---|---|---|---|---|
| | 2 views | 3 views | 5 views | 2 views | 3 views | 5 views | 2 views | 3 views | 5 views | 2 views | 3 views | 5 views | 2 views | 3 views | 5 views |
| cinema | 0.141 | 0.052 | 0.041 | 0.067 | 0.040 | 0.050 | 0.023 | 0.017 | 0.015 | 0.018 | 0.005 | 0.006 | 0.006 | 0.006 | 0.007 |
| boots | 0.034 | 0.076 | 0.048 | 0.029 | 0.023 | 0.022 | 0.017 | 0.014 | 0.011 | 0.013 | 0.003 | 0.003 | 0.001 | 0.002 | 0.002 |
| baskets | 0.138 | 0.141 | 0.040 | 0.038 | 0.033 | 0.029 | 0.015 | 0.017 | 0.004 | 0.016 | 0.008 | 0.003 | 0.008 | 0.008 | 0.026 |
| carving | 0.029 | 0.032 | 0.031 | 0.031 | 0.024 | 0.026 | 0.008 | 0.011 | 0.014 | 0.009 | 0.004 | 0.003 | 0.006 | 0.005 | 0.006 |
| chef | 0.199 | 0.100 | 0.124 | 0.062 | 0.056 | 0.055 | 0.023 | 0.020 | 0.025 | 0.017 | 0.011 | 0.009 | 0.003 | 0.006 | 0.006 |
| food | 0.146 | 0.057 | 0.035 | 0.058 | 0.041 | 0.035 | 0.017 | 0.013 | 0.014 | 0.013 | 0.006 | 0.005 | 0.006 | 0.007 | 0.016 |
| average | 0.115 | 0.076 | 0.053 | 0.048 | 0.036 | 0.036 | 0.017 | 0.015 | 0.014 | 0.014 | 0.006 | 0.005 | 0.005 | 0.006 | 0.010 |

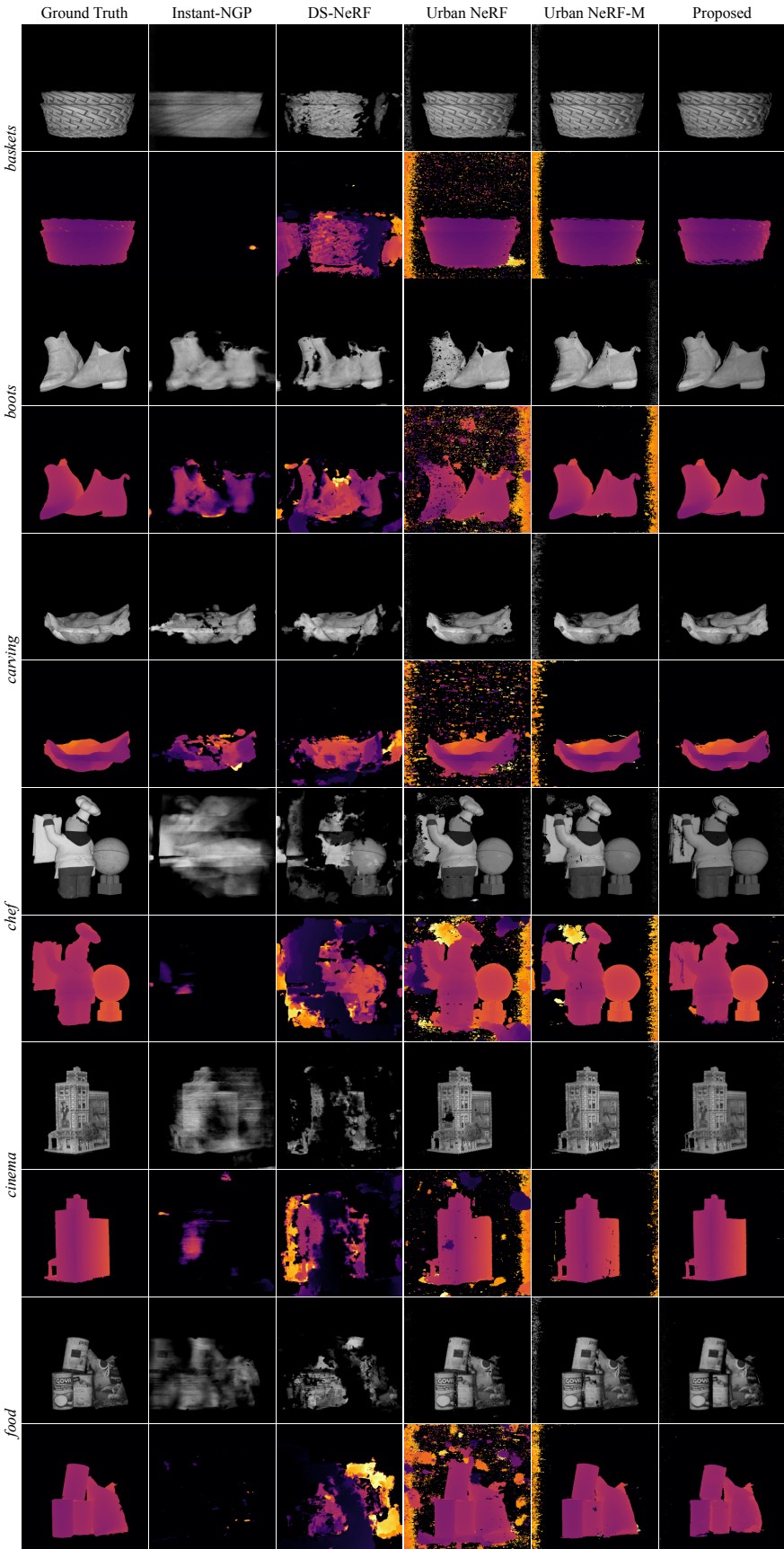

Figure S6: Rendered images and depths on the captured dataset for 2 views.

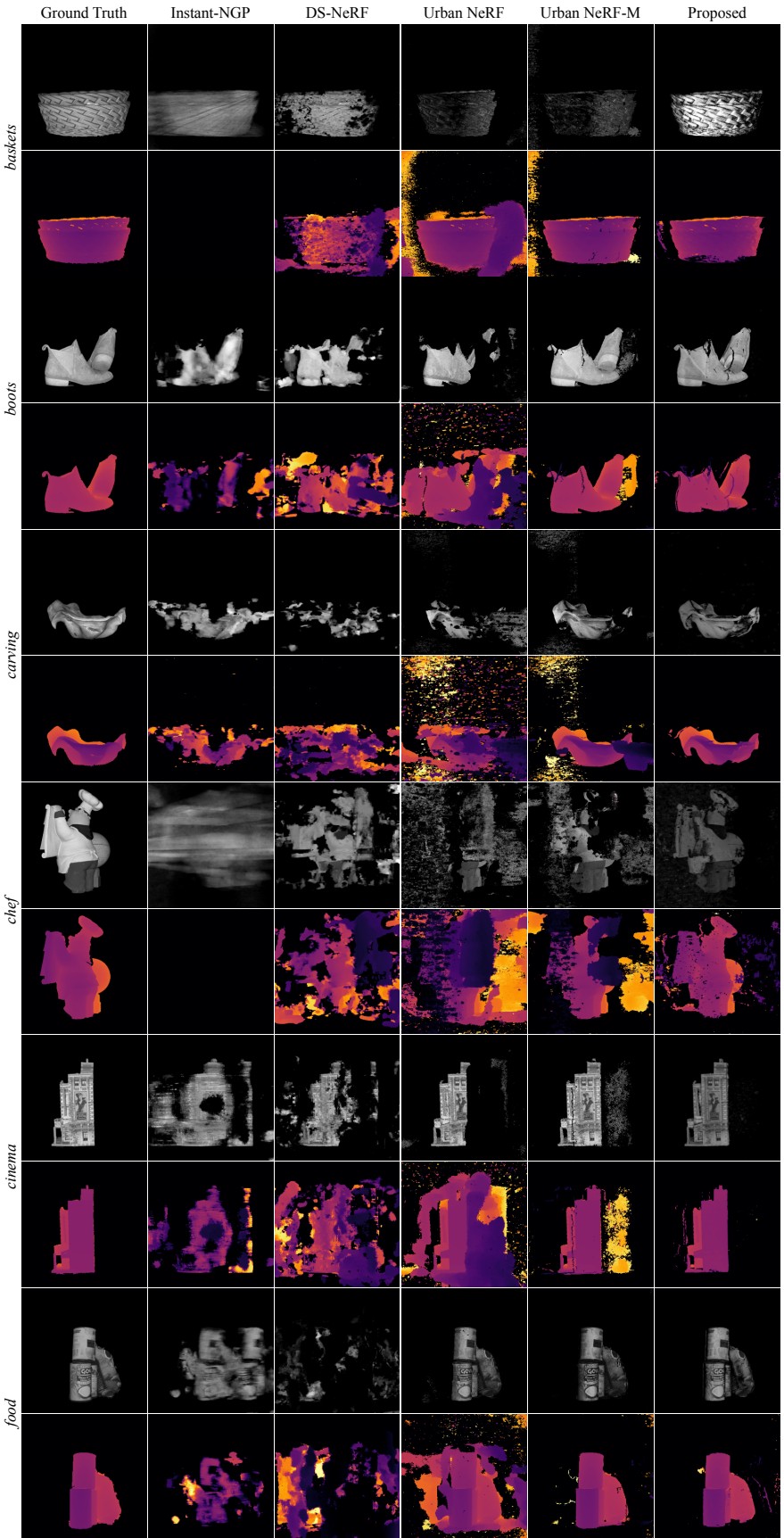

Figure S7: Rendered images and depths on the captured dataset for 3 views.

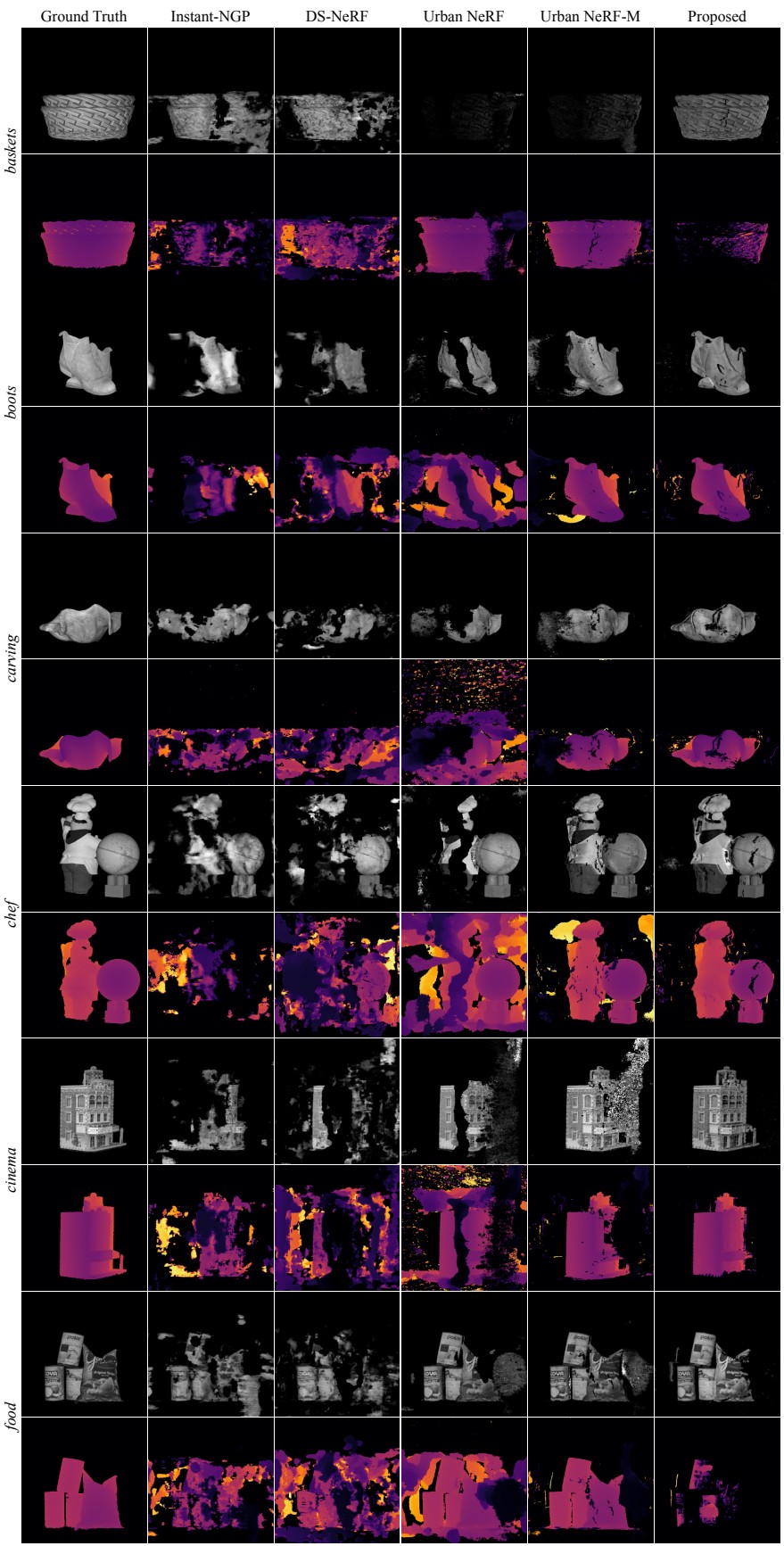

Figure S8: Rendered images and depths on the captured dataset for 5 views.