# OpenReview forum: "Transient Neural Radiance Fields for Lidar View Synthesis and 3D Reconstruction"
_NeurIPS.cc/2023/Conference — NeurIPS 2023 spotlight_

### Official Review · Reviewer_kaKw · 2023-06-19

**Soundness:** 3 good
**Presentation:** 3 good
**Contribution:** 3 good
**Rating:** 7
**Confidence:** 4

**Summary:**

The paper brings NeRFs to a new dimension of imaging at transient timescales enabling the rendering of transient imagery. It incorporates the underlying image formation model of the lidar and recovers improved geometry and conventional appearance when training on a few input viewpoints. The paper also proposes a first-of-its-kind dataset of simulated and captured transient multiview scans from a prototype single-photon lidar and tests the transient NeRF on the dataset.

**Strengths:**

The paper brings NeRFs to a new dimension of imaging at transient timescales and proposes a first-of-its-kind dataset of simulated and captured transient multiview scans from a prototype single-photon lidar. To deal with the new data type, HDR-Informed loss function and Space carving regularization are proposed to further improve the performance. The experiments are sound compared with other NeRF-based methods. The full paper has a clear structure and a clear narrative.

**Weaknesses:**

It only tests in a dark environment because of the single-photon Lidar. It is hardly used in broad application scenarios.

**Questions:**

Do you do the ablation study of HDR-Informed loss function and Space carving regularization? Can the method work under natural light?

---

> ### Author Rebuttal · Authors · 2023-08-09
>
> We thank the reviewer for the helpful feedback. Please see the above response to all reviewers where we address questions related to the ablation study of the HDR-informed loss and the space carving regularization. We discuss effects of natural lighting below.
>
> **kaKw: Scanning under natural light.** Our hardware setup is relatively robust to natural light because we place a laser-line spectral filter (Thorlabs FL532-10) in front of the SPAD, which attenuates ambient illumination by a factor of 10,000. Under indoor lighting, we observe roughly 300–3,000 photon counts per second, depending on the albedo of the target, which is <2% of the detected laser photons (~150,000 counts per second). However, operation in much brighter environments (e.g., outdoors under direct sunlight) would result in non-negligible background counts. In that case, it is possible to use other laser wavelengths (e.g., 1550 nm), where sunlight is heavily attenuated because of absorption by the atmosphere. We will note this in the paper.

---

> ### Comment · Reviewer_kaKw · 2023-08-13
>
> Thank you for the response. I am satisfied.

---

### Official Review · Reviewer_vJ74 · 2023-07-04

**Soundness:** 4 excellent
**Presentation:** 4 excellent
**Contribution:** 4 excellent
**Rating:** 9
**Confidence:** 4

**Summary:**

The paper describes a variant of NeRF specially designed for the image formation model of a time-of-flight imaging sensor. In contrast to earlier works using ToF sensors in a neural radiance field setting, the method is not designed to merely take depth maps or point clouds from the ToF sensors as input. Instead, the formulation goes one level deeper and presents a NeRF and image formation model on the basis of the time resolved photon count histograms that are used as basis for the final depth measurement by the ToF camera. To this end, the authors propose a time-resolved version of the volume rendering equation for image formation, as well as adapted HDR aware loss functions for training the neural radiance fields on photon histograms. The approach is tested on simulated data as well as real data captured with a ToF prototype setup. The authors show that in particular in a sparse view setting, the new photon count based formulation has advantages over depth-supervised or image-based NeRFs. Overall, I think this is a strong paper showing very innovative ideas. I enjoyed reading it.

**Strengths:**

Overall, I think this is a strong paper. It introduces quite a few clever ideas on how to adapt the neural radiance fields concept to the image formation model of a time-of-flight camera. This ranges from adapting the image formation model to a time-resolved version, to introducing a HDR aware loss and a space-carving regularization tailored to the peculiarities of the imaging properties of the employed sensor. To my knowledge, this is the first paper approaching neural radiance fields in this way.  As the authors suggest in their discussion the formulation on the basis of time resolved photon histograms may enable additional improvements in 3D reconstruction, e.g. of complex shapes, material properties etc.

**Weaknesses:**

Not much to say here. One point that could have been discussed more is from what number of viewpoints on the traditional or depth-supervised approaches catch up. At the moment, advantages of the approach are mostly shown in the sparse view setting, which is fine and showcasing an advantage of the approach.

**Questions:**

see the weaknesses section

One issue with time-of-flight sensors in general is their often non-trivial noise characteristics. Would explicitly incorporating the noise model into the formulation help here ?

**Limitations:**

The discussion in the paper is adequate.

---

> ### Author Rebuttal · Authors · 2023-08-09
>
> We thank the reviewer for the helpful feedback. Please see the above response to all reviewers where we discuss training on more viewpoints. We also discuss noise modeling below.
>
> **vJ74: Forward model choices–noise modeling**. As we note in the paper, the SPADs used in our hardware prototype follow a Poisson noise model (Eq. 2). Previously, we experimented with a loss function based on the corresponding negative log-likelihood (see, e.g., the formulation in O’Toole et al. [13]). Empirically, we found the L1 loss of the log transients (Eq. 5) to work well while also being simple to implement. However, we think incorporating additional reconstruction priors motivated by Poisson statistics could be a very promising direction for future work (see, e.g., Rapp and Goyal [17]).

---

### Official Review · Reviewer_Zi1e · 2023-07-04

**Soundness:** 3 good
**Presentation:** 4 excellent
**Contribution:** 2 fair
**Rating:** 5
**Confidence:** 4

**Summary:**

NeRF is a method that has recently become popular for view synthesis. It allows estimation of a 3D scene density and 5D radiance map using a few intensity images and their camera poses. These can then be used to render intensity and depth images from any novel 2D view.

This paper extends NeRF to allow estimation of scene density and radiance from “scene transients” instead of intensity/depth images. Previous methods have tried to incorporate depth information into NeRF in the form of point clouds, but the proposed method directly uses raw scene transients (histogram of photon counts) captured by a SPAD sensor. This method can then be used to render scene transients given a novel 2D view.

The authors' main contribution is to extend the volume rendering equation of NeRF to produce time resolved measurement instead of integrated over time. Additionally their training/rendering takes into account the shape of the laser pulse, spatial footprint of the laser spot etc. And lastly they add modifications to the NeRF objective to ensure their method works well for the high dynamic range SPAD data (predict radiances in the double log space, and apply loss in the log space).
The authors also apply a regularization (space carving) to encourage the density estimates to be sparse (similar to what other NeRF variants have done e.g urban NeRF).

The authors also capture a real world multiview LiDAR dataset with their hardware prototype. Using this dataset and another simulation based dataset, they provide quantitative and qualitative comparison of their method with other baselines for intensity and depth reconstruction from novel views. The results from simulation data show their method’s superior performance. The results from captured data also show that their method is superior, but by a smaller margin (which they attribute to alignment imperfections in the captured dataset).

**Strengths:**

1. This is a nice extension to the NeRF setup that allows directly leveraging the raw lidar measurements to learn a 5D representation instead of learning from intensity/depth images which won't contain all the information available in the raw measurements. The improvement in estimation accuracy is evident in the results/experiments shown.
2. The paper is written very well, with good attention to details (e.g. modeling assumptions, experiment setup). Background work is explained well.
3. Authors perform exhaustive experiments with baseline methods using both simulated and captured data, and results look impressive.


**Weaknesses:**

From a novelty of algorithm standpoint, the contributions of the paper are limited. From what I understood, It is a straightforward application of the NeRF framework to a new domain, i.e, the authors represent the scene density/radiance just like in NeRF, the only difference is that they modify the rendering logic to produce 3D histograms instead of 2D images. The image formation model of SPADs (which is used in the rendering equation) is also not novel.

**Questions:**

1. You mention that you model the spatial footprint of the laser/sensor spot when rendering the transients. Is that necessary? Does performance deteriorate if you just assume an ideal spot?
2. I'm not convinced about the utility of rendering raw lidar measurements from novel viewpoints. Other than generating simulation data, are there any practical applications?

**Limitations:**

Yes

---

> ### Author Rebuttal · Authors · 2023-08-09
>
> We thank the reviewer for the helpful feedback. Please see the above response to all reviewers where we address questions related to the spatial footprint of the laser/sensor spot. We address other questions related to novelty and applications below.
>
> **Zi1e: Novelty.** Our method is the first to apply a NeRF-type approach to multiview rendering of single-photon lidar data. Algorithmically, we extended the NeRF method to properly model a single-photon lidar system, including time-resolved rendering and modeling the temporal response of the system (i.e., laser pulse width and sensor jitter) as well as the footprint of the laser and detector. Further, we demonstrated our approach in practice with a first-of-its kind multiview single-photon lidar setup and captured dataset. We believe the novel hardware prototype and dataset will be appreciated by the community and inspire follow-on work.
>
> **Zi1e: Transient rendering applications**. One compelling application of the method could be generating novel views of lidar measurements for downstream tasks in training autonomous driving or robotics. We also believe that our work is an important first step for the task of transient neural rendering, which could enable advances in material segmentation, recovery of reflectance functions (e.g., BSSRDFs) from sparse views, non-line-of-sight imaging, and free-viewpoint rendering of light propagation and optical phenomena.

---

> > ### Comment · Reviewer_Zi1e · 2023-08-17
> >
> > Thanks for your response. It would be great if you could include the motivation for spatial footprint (the depth discontinuity failure case that you mentioned) in the final revision.

---

> > > ### Author Response · Authors · 2023-08-17
> > >
> > > Thank you, reviewer Zi1e. We will include the above motivation in the paper as requested.

---

### Official Review · Reviewer_tsWB · 2023-07-06

**Soundness:** 3 good
**Presentation:** 3 good
**Contribution:** 3 good
**Rating:** 4
**Confidence:** 4

**Summary:**

This work proposes a novel-view synthesis method for active sensors (single-photon LiDAR sensors) based on the Neural Radiance Field formulation. To this end, the volume rendering formulation for active sensors is derived, properly taking into account the measurement formation process (two-way distance, intensity fall-off) of LiDAR sensors. Along with that, a new loss functions for empty-space carving is proposed. The proposed method is implemented based on the INGP NeRF model and evaluated on both synthetic dataset of transient multiview scans (one of the contributions of this work), as well as on real-world data that was captured using a prototype single-photon LiDAR system. In these experiments, the proposed method outperforms SoTA NeRF methods (also with depth supervision) across all metrics.

**Strengths:**

- Original and important problem formulation. Novel-view synthesis for active sensors is an under researched, but a very important problem.
- A volume rendering formulation for active sensors that properly takes (at least some of the) the properties of the measurement formation process into account. These include, two-way path when computing the transmittance, intensity falloff, beam divergence.
- Simulated dataset of transient multiview scans that will be made publicly available (along with the scripts used to generate it)
- A dataset of real-world captures that were acquired using a prototype single-photon LiDAR which, if I understand correctly, will also be made publicly available

**Weaknesses:**

There are two main weaknesses in my view:
- **Clarity**: The clarity of the paper could in my opinion be improved, and the main confusion stems from including the RGB images from simulated data in figure 1 and using **c** to denote number of photons (?) in Figure 2. I might have misunderstood something, but if I am not wrong, the actual LiDAR system can only measure the photon count and depth (time-of-flight) and the volumetric rendering formulation should be tied to that. However, in the Equation 3 the radiance is a vector quantity (RGB color?)  same as in L150? The simulated data seems to be full RGB (what is the reason for this?), does that mean that for the simulated data the evaluation is actually similar to the original NeRF (PSNR and LPIPS computed over the RGB images?).

 - **Experimental evaluation**:  In my perception, the experimental evaluation seems somewhat biased. The proposed method is compared to the following baselines *INGP*, *DS-NeRF* and *UrbanNeRF* which are supervised through color/intensity supervision (and *DS-NeRF* and *UrbanNeRF* also using the depth). However, the depth supervision only constrains the integral over the density and the supervision signal might even be very noisy (log-matched filter in case of real world data are usually very noisy). On the other hand, the proposed method is supervised with the full histograms per each ray (before integration) which provide additional supervision and helps to constrain the empty space. This is especially important in the low-view setting that was for some reason selected in the evaluation (2-5 views on synthetic data?). I am wondering why the low-view setting was selected for simulated data? In my opinion this pronounces the bias as it emphasizes the differences in the supervision signal between the methods

**Questions:**

- What exactly does **c** denote in Equation 2 and L150?
- I like that the formulation tries to follow the measurement formation process, but currently only the intensity fall-off and round-trip are considered. Would it make sense to add at least the effects of the incidence angle?
- Similar to the comment above: why does simulated data simulate RGB and why is the evaluation considering low-view setting?
- I also like that the beam divergence is modeled through sampling multiple rays (L180), but it would be good to clarify at which point in the volume rendering formulation the contributions are averaged?
- The depth of the proposed methods is obtained as the argmax across the histogram, however for the baselines the integral along the ray is used. Does that yield better results as taking the argmax of the volume density profile?


**Limitations:**

The limitations are discussed in the conclusion section, while broader societal impacts are not (but I also don't believe that there are any significant ones).

---

> ### Author Rebuttal · Authors · 2023-08-09
>
> We thank the reviewer for the helpful feedback. Please see the above response to all reviewers where we address questions related to the experimental evaluation, effects of incidence angle, consideration of the low-view setting, and the depth evaluation. We address other questions below.
>
> **tsWB: Will the dataset be made public?** Yes, both the captured and simulated datasets will be made public upon publication of the paper.
>
> **tsWB: Clarification on dimensionality of symbol “$\mathbf{c}$” in rendering equations.** The lidar system measures photon count histograms. We simulated RGB photon count histograms, which could be captured in practice using an RGB laser. However, in our experimental setup, we only had access to a single-wavelength picosecond laser (green; 532 nm), and so our captured results have a single color channel. Hence, in our model, the dimension for $\mathbf{c}$ (i.e., the radiance output by the neural representation at each point along a ray) is different when applied to simulated and captured data. Specifically, in the simulations we use $\mathbf{c}\in\mathbb{R}^3$, whereas in the captured data $\mathbf{c}\in\mathbb{R}^1$.
>
> **tsWB: Clarification on evaluation metrics**. For evaluation on simulated data, we chose to use three color channels (RGB) and selected the evaluation metrics (LPIPS, PSNR) to be consistent with previous work (Instant NGP, DS-NeRF, Urban NeRF). To generate 2D images from the ground truth photon count histograms (or histograms rendered by the proposed method), we integrate the histograms over the time dimension and apply normalization and gamma correction (L259–262, L270–282). We also evaluated the depth estimates produced by our method as described on L182–185.
>
> **tsWB: Clarification on spatial footprint modeling.** The weighted summation is done as the last rendering step by averaging the corresponding time bins within the photon count histograms produced by each ray for a given pixel. We will clarify this in the revision.

---

> > ### Comment · Reviewer_tsWB · 2023-08-13
> > **Response to the rebuttal**
> >
> > I would like to thank the authors for providing a detailed response. Some of my concerns were addressed, but I would still like to note a couple of things.
> >
> > **Clarification on dimensionality of symbol in rendering equations.**
> >
> > It would be good to make this clear in the paper (for other scalars you use non-bold font, so this might be confusing to others as well). If I understand things correctly (it might be that I don't, I am not an expert on the photon lasers), the photon count output of your network is strongly correlated with the density (module the differences in material and translucence). It would be good to discuss this in the paper and provide some histograms of both density and photon count ($\textbf{c}$) outputs of your network.
> >
> > **Comparison to baselines.**
> > I would respectfully disagree that the comparison to the baselines is completely apples-to-apples. While the raw data is indeed the same, the baselines methods only have access to the data after the log-matched filter, which reduces a histogram to a single value representation (obviously a lot of information is lost here). The histogram data is especially useful in the sparse view setting as it provides **spatial supervision** that helps to constrain the solution (empty space supervision) which other methods don't have access to. I would even go as far as to argue that the main insight ***raw measurements provide better supervision*** was "known/suspected" before. Baseline methods do not use spatial data as it is simply not available by most consumer grade LiDAR systems (full-waveform LiDAR are rare in robotics, AV and other typical use cases). In fact, due to the missing spatial data, previous methods try to reason about it using heuristic observations, e.g. UrbanRadianceFields introduce line-of-sight priors about empty space and center a Gaussian at the measured depth to obtain additional supervision.
> >
> > Do not get me wrong, I think that this work is good, and the insights are interesting, but I would wish that they would be presented in a bit more "humble" (for the lack of better term) way and also discuss the availability of such data to prior methods and its effect on the evaluation. I am convinced that the previous works would also try to use full histogram data if it would be available to them (incorporating manually defined spatial priors based supports this). Nevertheless, the proposed formulation is interesting and might spark new ideas in the future.

---

> > > ### Author Response · Authors · 2023-08-13
> > >
> > > We thank the reviewer for taking the time to respond.
> > >
> > >
> > > **Clarification on dimensionality of symbol “$\mathbf{c}$” in rendering equations.** We will clarify the dimensionality of $\mathbf{c}$ (the radiance predicted by the network) in the paper. Indeed, the time-resolved output of the network depends on both the density and the radiance, and will show some plots in the revision to visualize how these quantities compare to the rendered output along the ray.
> > >
> > >
> > > **Comparison to baselines.** We agree with the reviewer that our method differs from the baselines in terms of using point cloud data vs. photon count histograms. In the rebuttal we intended the “apples-to-apples” wording to emphasize that the evaluation of the proposed method uses the same photon count histograms as are used to estimate the point clouds used for the baselines. Since previous methods do not have access to histogram data, we used the “Urban-NeRF-M” baseline to explore performance of point cloud-based methods with additional spatial priors—here we augmented previous work with ground truth segmentation masks to facilitate space carving.
> > >
> > >
> > > While it’s perhaps intuitive that using the raw measurements should provide some improvement over using point cloud data alone, it was not obvious to us a priori how significant the benefits would be. Moreover, implementing time-resolved NeRF supervision using photon count histograms was not trivial and required accounting for additional factors (e.g., laser/sensor footprint, system temporal response, building the hardware prototype, etc.). Additionally, we believe the proposed work provides value because it quantifies the improvements from using raw lidar data in the context of NeRF reconstruction.
> > >
> > >
> > > The reviewer is also correct that most conventional lidars do not output full waveform data—while they typically use fast avalanche photodiodes to capture measurements, and they initially capture the full lidar waveform, the data are preprocessed to point cloud format before output. We will discuss the fact that raw lidar data were not readily available to previous methods. We hope that our dataset of multiview photon count histograms will provide more opportunities to investigate 3D reconstruction using raw lidar data.

---

> > > > ### Comment · Reviewer_tsWB · 2023-08-15
> > > >
> > > > Thank you for your response and clarifications.
> > > >
> > > > Indeed, clarifying the dimensionality of $\mathbf{c}$ and including the plots of both density and radiance will help readers to get a better intuition.
> > > >
> > > > **Comparison to baselines**:
> > > >
> > > > I fully agree that the level of improvement was not clear before, and I can see how the implementation (making it work) was not trivial. The discussion you provided and intuition you provided above is exactly what I would like to see in the paper. It was not my intention to diminish the contributions of this work, I was just hoping that they can be put into the perspective better.
> > > >
> > > > If the authors intent to include the discussion above in the paper, I am happy to increase my rating.

---

> > > > > ### Author Response · Authors · 2023-08-15
> > > > >
> > > > > Thank you, reviewer tsWB. We will include the above discussion in the paper as requested.

---

### Author Rebuttal · Authors · 2023-08-09

We thank all reviewers for their helpful suggestions and insightful questions.  Our work brings NeRFs to a new dimension of imaging at transient timescales, enabling rendering of photon count histograms from novel views for the first time. Further, we show that supervision on photon count histograms enables improved reconstruction of geometry and novel view synthesis compared to point cloud-based supervision when training on few input viewpoints. Our first-of-its-kind multiview dataset of photon count histograms (including both simulated and captured data) will be made publicly available upon publication.

We address most reviewer concerns in this shared section of the rebuttal and respond to other individual concerns as separate responses to each reviewer. Please see the attached rebuttal PDF for Tables R1–R6.

**Adjustment to simulated results.** After submission, we noticed a slight inconsistency with the pre-processing of the simulated dataset which we have now corrected. Specifically, we created the RGB images used for training by summing the simulated photon count histograms across the time dimension and normalizing by the maximum value _per view\._ We reprocessed the simulated results to use normalization _across all views_ (consistent with the captured results, L230-231). We include the updated results in Table R1 of the rebuttal PDF (compare to Table 1 of the paper). The overall trends and qualitative results are still consistent with what we previously reported, and our method still outperforms all baselines in terms of novel view synthesis and depth estimation. We will use these results in the revision unless reviewers object. Tables R2 and R3 also use normalization across all views.

**tsWB, vJ74: Number of views.** The paper focused on results for sparse views (i.e., 2, 3, and 5 views) since this is the regime where depth supervision improves the most over only using 2D supervision. We tried using 10 simulated training views sampled at equal angles around the Lego scene and our approach still outperforms the baselines when evaluated on the same test views (see Table R2). If reviewers request, we can include results with even more training views in the revision.

**tsWB: Clarification on experimental evaluation.** The reviewer notes that the depth supervision used by previous methods (DS-NeRF, Urban NeRF) only constrains the integral of the density based on point cloud data (though Urban NeRF also includes point cloud-based space carving losses, see their Eq. 15).

It turns out that supervision using the raw lidar data (i.e., photon count histograms) provides better performance, especially for reconstruction from few input views; **this is a key insight of the paper**. The evaluation uses fair, apples-to-apples comparisons based on the same input photon count histograms. That is, for DS-NeRF and Urban NeRF, we estimate point clouds by applying the constrained maximum likelihood estimate (i.e., log-matched filter) to the photon count histograms. The proposed method uses those same histograms, and we show that supervision with raw data outperforms using pre-processed point clouds in the sparse-view setting considered by DS-NeRF and Urban NeRF. **Importantly,** **this result improves over the standard practice of preprocessing raw lidar data to point clouds.**

**tsWB: Depth evaluation.** DS-NeRF and Urban NeRF calculate depth differentiably using the integral along the ray (i.e., the expected ray termination distance) and incorporate this in their loss functions. Since the baseline methods are supervised with this depth estimate, we opted to use the same approach for evaluation.

For completeness, we provide an updated version of the L1 depth in Table R3 calculated for all methods using the maximum ray termination probability (i.e., using argmax; see L183). Baseline performance using argmax is indeed improved for most of the methods, though Transient NeRF still performs best.

**tsWB: Normals in forward model**. Estimating the normals requires a noisy finite-difference operation on the predicted density values, which we found results in speckle-like artifacts in the novel views. Instead, we model the effects of incidence angle by conditioning the neural representation on view direction. See Table R4, which ablates using the normals to add cosine factors to the rendering equation (Eq. 3) for the Cinema scene.

**Zi1e: Ablation–laser spatial footprint.** When the laser spot and sensor footprint pass over a depth discontinuity, we observe two peaks in the resulting photon count histogram (corresponding to the two depths across the discontinuity). Assuming an ideal spot (i.e., using a single ray) completely fails to model this effect. Training and rendering the Lego scene (which has many depth discontinuities) with the ideal spot model results in much worse performance across novel views (see Table R5).

**kaKw: Ablation–HDR-informed loss function.** We provide an ablation study of the HDR-informed loss function in Table R6 on the Chef and Food scenes. Incorporating this loss provides some improvement by preventing very bright regions from dominating the loss. Moreover, in the case of 5 input views, performance without the HDR-informed loss drops due to specular highlights appearing in one view, but not an overlapping nearby training view (e.g., on the globe in the Chef scene or the bag of chips in the Food scene). Without the HDR-informed loss, the network has difficulty modeling the large variation in radiance between views, and so the optimization produces spurious patches of density to model this view-dependent effect (despite regularization with the space carving loss).

**kaKw: Ablation–space carving.** An ablation study of the space carving regularization on the Cinema scene is in Supp. Table 2 (reproduced in Table R4). Without space carving regularization we find that spurious clouds of density appear in empty regions, worsening performance across most metrics.

---

### Decision · Program_Chairs · 2023-09-21

**Decision:**

Accept (spotlight)

**Comment:**

The reviewers are more or less unanimously in favour of acceptance tsWB has not upgraded to a positive score, but in the discussion they have made clear that they find the paper interesting and relevant and that they would increase their rating if the authors include content from the discussion - which they have promised. The AC agrees that the question of supervision by photon histograms vs. discrete range measurements should be elaborated in the paper, please make sure to do this for the final version.